# SPURIOUS REWARDS: RETHINKING TRAINING SIGNALS IN RLVR

## ABSTRACT

We show that reinforcement learning with verifiable rewards (RLVR) can elicit strong mathematical reasoning in certain models even with *spurious rewards* that have little, no, or even negative correlation with the correct answer. For example, RLVR improves MATH-500 performance for Qwen2.5-Math-7B in absolute points by 21.4% (random reward), 13.8% (format reward), 24.1% (incorrect label), 26.0% (1-shot RL), and 27.1% (majority voting)—nearly matching the 29.1% gained with ground truth rewards. However, the spurious rewards that work for Qwen often fail to yield gains with other model families like Llama3 or OLMo2. In particular, we find code reasoning—thinking in code without actual code execution—to be a distinctive Qwen2.5-Math behavior that becomes significantly more frequent after RLVR, from 65% to over 90%, even with spurious rewards. Overall, we hypothesize that, given the lack of useful reward signal, RLVR must somehow be surfacing useful reasoning representations learned during pretraining, although the exact mechanism remains a topic for future work. We suggest that future RLVR research should possibly be validated on diverse models rather than a single de facto choice, as we show that it is easy to get significant performance gains on Qwen models even with completely spurious reward signals.

## 1 INTRODUCTION

Reinforcement learning with verifiable rewards (RLVR) is highly effective in enhancing language model reasoning (Lambert et al., 2024; DeepSeek-Math, 2024; Zeng et al., 2025; Luo et al., 2025b). We show, counterintuitively, that RLVR can improve mathematical reasoning even with weak or flawed *spurious rewards* when applied to Qwen2.5-Math models (Yang et al., 2024a;b), a popular and performant model family used in the RLVR literature (Hu et al., 2025a; Yang et al., 2025; Wang et al., 2024; Guan et al., 2025; Zeng et al., 2025) (§2). For example, using incorrect labels for training results in 24.1% absolute accuracy gain on MATH-500, while using format or random rewards result in 13.8% and 21.4% gains, respectively. Strikingly, these spurious-reward gains are even comparable to the 29.1% gain from training on ground truth. We observe similar trends on more challenging math benchmarks such as AMC and AIME. Overall, our findings suggest that we do not yet fully understand the exact mechanisms by which RLVR improves performance, and that in many cases it may be somehow exposing innate model abilities learned during pretraining, in addition to whatever reward signal it is getting.

We further present an extensive experimental study to measure the improvements from weak and spurious rewards with a cross-model analysis (§3). For model families not specifically optimized for mathematical reasoning during pretraining—including Qwen2.5 (Yang et al., 2024b), OLMo2 (OLMo et al., 2024), and Llama3 (Dubey et al., 2024) variants—we observe a critical divergence: OLMo and Llama models (i.e. non-Qwen models) show minimal improvement or even become worse after training on spurious rewards, strongly suggesting that differences in pretraining at least in part explain the difference in RLVR.

To help explain this discrepancy, we also analyze what reasoning patterns that RLVR is learning to favor in these cases (§4). In particular, we find a majority of Qwen2.5-Math-7B answers on MATH-500 contain reasoning chains expressed in Python—a behavior we call *code reasoning*—despite having no access to code execution. Code reasoning is highly predictive of overall performance; answers with it have an accuracy of 60.9%, much higher than without (28.0% accuracy). Code reasoning also correlates with MATH-500 accuracy over the course of RLVR training. Both metrics increase consistently during training with any spurious reward, leading to $\sim 90\%$ or higher code

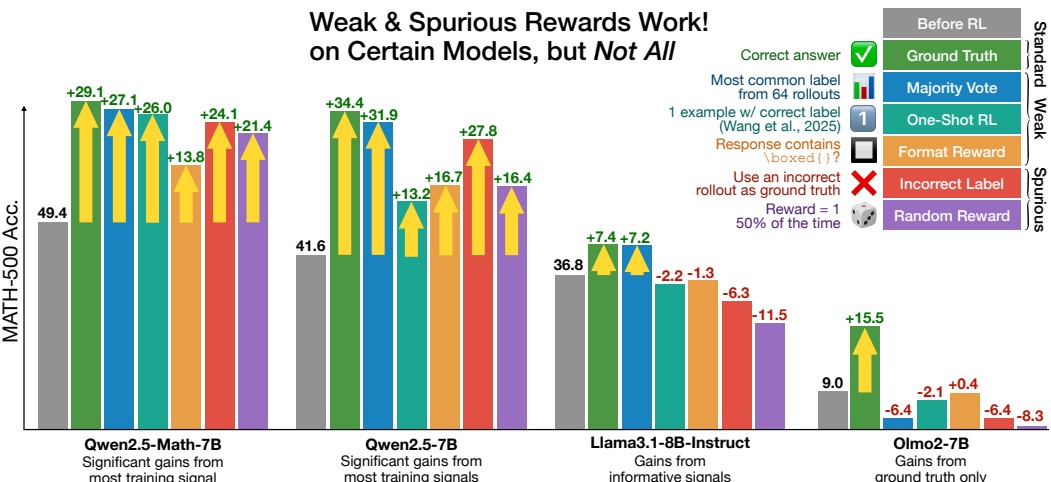

Figure 1: MATH-500 accuracy after 300 steps of RLVR on various training signals. We show that even "spurious rewards" (e.g., rewarding *incorrect* labels or with completely random rewards) can yield strong MATH-500 gains on Qwen models. Notably, these reward signals do not work for other models like Llama3.1-8B-Instruct and OLMo2-7B, which have different reasoning priors.

frequency after training. Based on this observation, we further hypothesize that intervening with other methods that increase code frequency should similarly increase test performance. Our experiments validate this hypothesis: we design prompt-based and RL-based code reasoning elicitation methods to increase code reasoning; all such methods significantly increase Qwen2.5-Math-7B's performance. We show that these results are robust to prompt variations, despite the initial model performance being sensitive to prompts in sometimes unexpected ways. We conclude by discussing potential explanations for why spurious rewards can still work.

Our findings not only open new questions but also have practical implications. We should generally be more aware that reasoning patterns instilled during pretraining heavily impact the behavior of downstream RLVR training, with code reasoning ability standing out in our study. Qwen models, with open weights and high performance on reasoning tasks, have become the de facto choice for RLVR research in the open-source community—a range of recent research on RLVR drew conclusions on Qwen2.5-Math-7B-centric experiments (Zuo et al., 2025; Zhao et al., 2025c; Wang et al., 2024; Xie et al., 2025; Hu et al., 2025a; Zhang et al., 2025; Shafayat et al., 2025; Prabhudesai et al., 2025; Gao et al., 2025; Wang et al., 2025a). However, we show that it is easy to get significant performance improvements on Qwen models even with completely spurious reward signals. Thus, we suggest that future RLVR research should possibly be confirmed on other models.

## 2 SPURIOUS REWARDS YIELD SIGNIFICANT RLVR GAINS

We design a progression of reward functions to replace the standard ground-truth reward: *weak rewards* (majority vote reward and format reward) and *spurious rewards* (random reward and incorrect reward). Remarkably, we find that all weak and spurious rewards suffice for RLVR to significantly improve the math performance of Qwen2.5-Math, a popular starting point for RLVR training.

### 2.1 EXPERIMENTAL SETUP

Following recent RLVR work (Wang et al., 2025b; Zuo et al., 2025; Zeng et al., 2025), we use GRPO (DeepSeek-Math, 2024) to finetune Qwen2.5-Math models (Yang et al., 2024a). The standard RLVR approach uses a dataset of questions paired with ground truth labels. During training, model rollouts are given a binary (0-1) reward based on whether the generated answer is verifiably correct. We replace this ground truth-based reward with a variety of increasingly spurious binary 0-1 reward functions that do not require access to ground truth labels. We design these alternative rewards to investigate the limits of how little supervision is needed for effective RLVR training. We train on DeepScaleR data (Luo et al., 2025b) with our rewards; all other experimental details are kept constant.

In the main paper, we evaluate performance as pass@1 and average@8 accuracy on two standard math reasoning benchmarks: MATH-500 (Hendrycks et al., 2021) and AMC (Li et al., 2024), respectively.

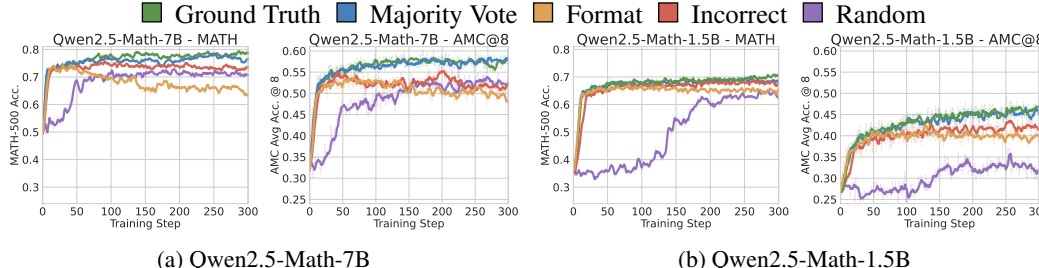

(a) Qwen2.5-Math-7B          (b) Qwen2.5-Math-1.5B

Figure 2: Model performance on MATH and AMC with varied training rewards smoothed over window size of 10 (dotted lines are unsmoothed values). We report pass@1 for MATH and average@8 for AMC. Both Qwen2.5-Math-7B and Qwen2.5-Math-1.5B significantly improve after RLVR on a range of reward signals from meaningful to spurious. We note that the random reward converges slower than the other spurious rewards, but the fact that it leads to significant gains at all is surprising.

See Appendix D for additional results on AIME 2024 and 2025. Following the default evaluation setup in the popular RL framework OpenRLHF (Hu et al., 2024), we use the default chat template for Qwen2.5-Math and all instruct models and chat template provided by Olmo (OLMo et al., 2024) for other base models in our main experiments. Additional analysis on the effect of different prompts can be found in Appendix I, where we show Qwen2.5-Math-7B is very sensitive to prompts—even a task-irrelevant prompt (which we name *spurious prompt*) can sometimes result in high initial performance. See Appendix A for full details of our training and evaluation setup.

### 2.2 STANDARD TO WEAK TO SPURIOUS REWARDS

We consider the following rewards:

1. **Ground Truth Rewards:** To establish a baseline, we consider the standard RLVR approach (Lambert et al., 2024) of using ground truth labels to reward responses with verifiably correct answers. This setting serves as an upper bound for reward supervision quality.

2. **Majority Vote Rewards:** Instead of using ground truth labels for computing rewards, we use the model prior to RLVR training to pseudo-label the training set by selecting the majority answer from 64 sampled responses per prompt. These (potentially wrong) labels are then used to reward responses during standard online RLVR training.

3. **Format Rewards:** We further weaken the reward signal to disregard the responses' mathematical correctness altogether. We instead heuristically reward all responses containing at least one non-empty `\boxed{}` expression, regardless of the correctness of the enclosed answer. Including `\boxed{}` is specified in Qwen2.5-Math's system prompt; this reward incentivizes some degree of prompt following.

4. **Random Rewards:** We study whether providing *no* guidance in the rewarding process is sufficient to provide meaningful math performance gains. To do so, we assign rewards randomly. Given a fixed probability hyperparameter $\gamma$, all responses receive a reward of 1 with chance indicated by the parameter, and receive 0 otherwise. In our main experiments, we present $\gamma = 0.5$; in Appendix B, we show that using $\gamma \in \{0.001, 0.3, 0.7\}$ obtains similar improvements with varying convergence speed, and verify that $\gamma = 0$ results in no change as expected analytically (with $\gamma = 0$, loss is constant, and all gradients are zero).

5. **Incorrect Rewards:** We furthermore deliberately provide incorrect supervision and reward only incorrect answers. We first label all training data using majority voting and select the subset with incorrect labels for training, obtaining incorrect labels that are still probable outputs of the models. During training, we reward responses whose answers verifiably match these incorrect labels.

### 2.3 RESULTS

Figure 2 presents the performance of Qwen2.5-Math models after RLVR training with each reward function. Overall, all reward functions, even pathologically designed ones, lead to significant improvements in math performance within the first 50 steps across all benchmarks compared to the untuned baseline. One exception is that Qwen2.5-Math-1.5B sees gains with random rewards

much slower (after 100 steps) and less so on AMC (only 4.9%). Remarkably, performance gains from spurious rewards are often within a few points of the gain from RLVR with ground truth labels. For example, training with incorrect label reward yields 24.1% gains over Qwen2.5-Math-7B on MATH-500, compared to a 29.1% gain from RLVR with ground truth answers. Even random rewards—which by design provide pure noise in the rewarding process —still produce a 21.4% performance boost. We observe similar trends in AMC, where training on format, incorrect, or random rewards yields a gain of 13.8%, 24.1%, or 21.4%, respectively, approaching the $\sim 27\%$–29% improvement gained from training on majority voted and ground truth labels.

In Appendix D, we supplement results on AIME24 and AIME25. On AIME2024, format reward (+10.3%) approaches ground truth rewards (+15.3%), and spurious rewards (incorrect & random) still lead to high performance gains of 10.2% and 10.2% respectively on Qwen2.5-Math-7B. Ground truth labels show a clear advantage compared to other rewards on AIME2025, which contains questions written after the knowledge cutoff of all models we consider. Nonetheless, other rewards still lead to a -0.4% to 4.5% gain in performance.

Our findings with these simple rewards provide additional evidence for a nascent hypothesis in the literature: that RLVR, at least at the compute scales of open-source post-training pipelines (Lambert et al., 2024), does not teach models new reasoning capabilities, but instead triggers latent ones already present in the base model (Wang et al., 2025b; Liu et al., 2025; Gandhi et al., 2025; Yue et al., 2025; Shah et al., 2025; Choshen et al., 2020). Whereas prior work has hinted at this effect using limited-quantity ground-truth labels (Wang et al., 2024) or noisy labels (Zuo et al., 2025), our results push this idea to its limit: we show that even outright incorrect rewards or information-free rewards (i.e., random) can elicit performance gains in Qwen2.5-Math models. In the remainder of our paper, we show that this elicitation effect is model-dependent (§3), and trace the specific properties of Qwen2.5-Math models that could enable spurious rewards to induce this elicitation (§4).

## 3   (LACK OF) GENERALIZATION TO OTHER MODELS

Inspired by the unexpected effectiveness of spurious reward signals in improving the performance of Qwen2.5-Math models, we study whether these rewards generalize to training other models. We extend beyond the math-specialized Qwen2.5-Math models to include general-purpose variants (Qwen2.5-7B, Qwen2.5-1.5B (Yang et al., 2024b)), and two additional model families: (a) the widely-used Llama3.1-8B(-Instruct) and Llama3.2-3B(-Instruct) (Dubey et al., 2024), and (b) OLMo2-7B and OLMo2-7B-SFT (OLMo et al., 2024). OLMo2-7B-SFT is instruction-tuned from OLMo2-7B; we include both to better understand the impact of SFT training on RLVR training. Moreover, we are optimistic that OLMo's open training data will enable future works to better study the origins of any reasoning behaviors (or lack thereof) we later observe. We train these 8 additional models with the same setup and rewards as in Section 2. We report performance on MATH-500 in the main text of the paper and AMC in Appendix C where the same trends are observed. The AIME 2024 and 2025 benchmarks show similar but noisy trends (Appendix D).

**Spurious rewards can benefit Qwen2.5 models, but nearly always fail to improve non-Qwen models.**   Figure 3 offers two key takeaways. First, models within the same family generally exhibit similar trends. For instance, for both Qwen2.5 models, all non-random rewards—including the spurious incorrect reward—produce clear improvements in MATH-500 performance. Both Olmo models show relatively flat performance on spurious rewards and only show significant gains with ground truth reward training. We conjecture that this consistency in behavior among models within the same family arises from similarities in their pretraining data distributions; models from the same family likely exhibit similar behaviors prior to RLVR training. Additionally, we find that smaller models are less likely to benefit from spurious rewards, such as random rewards. We conjecture that larger models retain more knowledge from pretraining that our spurious rewards elicit. In Section 4, we will further explore how differences in pre-existing model behaviors impact the outcome of RLVR.

Secondly, we observe that reward signals which work well for one model family do not necessarily generalize to other model families. Even though the spurious incorrect reward and all weak rewards yield consistent gains on top of Qwen models, each weak or spurious reward fails to produce similar gains for at least one other model, often resulting in flat or even decreased performance. Overall, our results suggest that Qwen models are uniquely robust to reward signal strength. Lastly, we

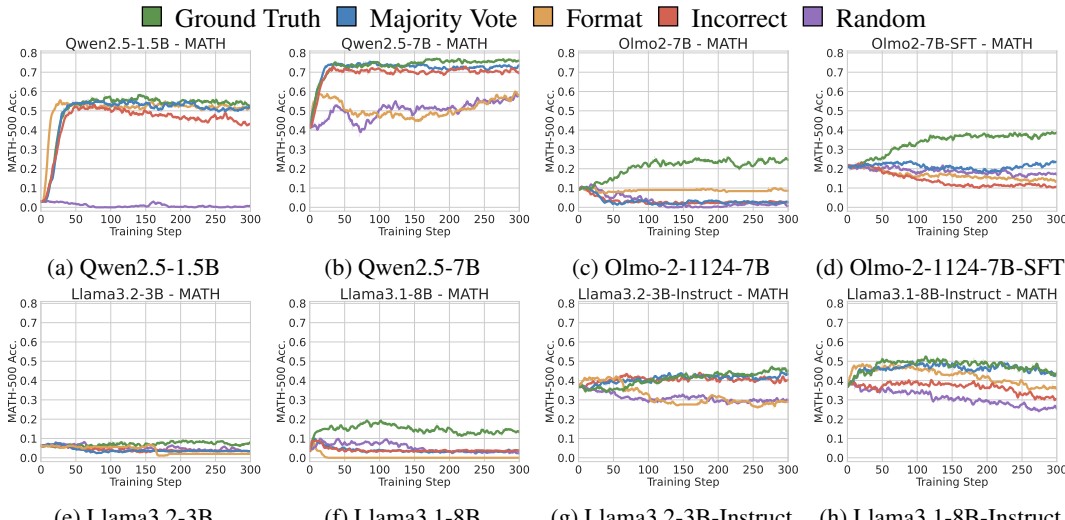

| | Ground Truth | | Majority Vote | | Format | | Incorrect | | Random |

(a) Qwen2.5-1.5B  (b) Qwen2.5-7B  (c) Olmo-2-1124-7B  (d) Olmo-2-1124-7B-SFT

(e) Llama3.2-3B  (f) Llama3.1-8B  (g) Llama3.2-3B-Instruct  (h) Llama3.1-8B-Instruct

Figure 3: Varying rewards across additional model classes on MATH-500. Spurious rewards remain effective on general-purpose Qwen2.5 models but generally fail to yield any gains on other model families. The performance improvements on non-Qwen2.5 models are substantially smaller compared to those observed in the Qwen2.5 family. Similar trends are observed on AMC in Appendix C.

present additional results for additional models that have already gone through RL post-training in Appendix J, showing that they have minimal improvements with nearly all rewards.

**Practical warning ⚠: Proposed RLVR reward signals should be tested on diverse models!** Many recent methods on RLVR for reasoning draw their conclusions primarily or exclusively from gains shown on Qwen models (non-exhaustively, Zuo et al. (2025); Zhao et al. (2025c); Wang et al. (2024); Xie et al. (2025); Hu et al. (2025a); Zhang et al. (2025)). As a case study, we experimented with recent work on (1) test time training (Zuo et al., 2025) and (2) one-shot RL (Wang et al., 2025b) (setup details in Appendix A.7). We find these methods exhibit a pattern similar to our results above: the proposed training signals yield strong improvements in the Qwen2.5-Math or Qwen2.5 models, matching the performance of the ground truth rewards (Figure 15 in Appendix E). However, these same signals often fail to yield performance gains on other model families. Our findings suggest that existing Qwen-centric RLVR research should possibly be further validated on non-Qwen models.

## 4 UNDERLYING DYNAMICS OF RLVR WITH SPURIOUS REWARDS

Previously, we demonstrated that different models can exhibit markedly different outcomes when trained with the same reward function. In this section, we investigate *why* such discrepancies occur. Broadly, we hypothesize that differences in RLVR training outcomes are due to differences in the specific reasoning strategies learned by each model during pretraining. In particular, some strategies may be readily elicited by RLVR, while others may be more difficult to surface or lacking altogether.

Below, we identify one such pre-existing strategy—generating code to assist in math reasoning—that Qwen-Math utilizes effectively, and other model families less so (§4.1). Tracing the prevalence of code reasoning over the course of RLVR training, we find strong evidence for our hypothesis (§4.2).

### 4.1 DIFFERENT MODELS EXHIBIT PRE-EXISTING REASONING STRATEGY DISCREPANCIES

We begin our analysis with a case study examining the reasoning traces generated by different models on the MATH-500 test set. Our investigation reveals a notable behavioral difference: Qwen2.5-Math-7B frequently incorporates Python code into its reasoning process (appearing in 65.0% of all responses), despite operating without access to a code execution environment. We term this phenomenon *code reasoning*. Figure 4 provides a representative example, with additional examples

and detailed analysis available in Appendix L. We use code reasoning *as an illuminating case study* for the subsequent analysis.

---

**MATH Question:**

What is the distance, in units, between the points $(2, -6)$ and $(-4, 3)$? Express your answer in simplest radical form.

---

**Qwen2.5-Math-7B Solution (correct):**

To find the distance between two points $(x_1, y_1)$ and $(x_2, y_2)$ in a Cartesian plane...
Let's break this down step-by-step and compute the result using Python.

```python
import math
...
# Calculate the distance using the distance formula
distance = math.sqrt(dx**2 + dy**2)
print(distance)
```

output: 10.816653826391969

...

Thus, the final answer is: $\boxed{3\sqrt{13}}$

---

Figure 4: Example of Qwen2.5-Math-7B's code reasoning (see Figure 22 for the complete response). The question is randomly picked from the MATH-500 test set. Note that both the code and the code execution result are autoregressively generated by Qwen2.5-Math-7B. **No external code interpreter was provided to the model.**

Notably, Qwen2.5-Math-7B's use of code is not merely superficial—code use is strongly predictive of answer correctness. As shown in Table 1, Qwen2.5-Math-7B achieves significantly higher accuracy on MATH prompts when code reasoning is used instead of natural language reasoning alone. Qwen2.5-Math-1.5B, pre-trained on the same corpus as Qwen2.5-Math-7B, exhibits similarly effective code reasoning. This pattern is not observed in other models, for which we categorize as "No-Code" or "Bad-Code." *No-Code* models—Llama models, Qwen2.5-1.5B, and OLMo2-7B—do not generate code at all, and consequently do not benefit from this reasoning strategy; *Bad-Code* models frequently attempt to use code reasoning—OLMo2-7B-SFT (98.0% of the time) and Qwen2.5-7B (92.2% of the time)—but code generation correlates with *worse* performance for these models. Thus, *effective* code reasoning is a unique pre-existing capability of the Qwen2.5-Math models before RLVR training.

We provide further analysis that hints at the origins of code reasoning of Qwen2.5-Math-7B models in Appendix L. Note that code reasoning is not used as a complete explanation: other behaviors can also be elicited easily and often correlate with performance—we briefly discuss another such behavior, generation without repetition, in Appendix G. In addition, we find the initial performance of Qwen2.5-Math-7B is sensitive to the prompts used for evaluation; we supplement a discussion on this impact in Appendix I.

### 4.2 RLVR WITH SPURIOUS REWARDS UPWEIGHT PRE-EXISTING REASONING STRATEGIES

Motivated by our observations above, we traced changes in the reasoning behavior of models throughout the course of RLVR training across two dimensions: (1) **Accuracy**: the average accuracy of the model on MATH-500, and (2) **Code reasoning frequency**: the percentage of model responses containing the string "python". We find that rewards that we employed in the paper, including spurious rewards such as the random and incorrect rewards, gain much of the accuracy on the Qwen2.5-Math and Qwen2.5 through eliciting the correct reasoning strategy.

**Performance is correlated with code reasoning frequency.** As shown in Figure 5, prior to RLVR training, Qwen2.5-Math-7B exhibits a high rate of 65.0% code reasoning solutions. After RLVR on all but random rewards, the frequency of code reasoning quickly increases to around 90% in the first

Table 1: We track the percentage of model-generated MATH-500 responses that contain Python code before RL training, as well as their accuracy on (1) responses with code and (2) responses with only natural language. Overall, Qwen2.5-Math models achieve higher performance when using code than when not, while other models do not benefit from code reasoning. The unlisted models (Qwen2.5-1.5B, OLMo2-7B, Llama3.1-8B-Instruct, Llama3.2-3B-Instruct) *never* generated code.

| Model | Qwen2.5-Math-7B | Qwen2.5-Math-1.5B | Qwen2.5-7B | OLMo2-7B-SFT |
|---|---|---|---|---|
| **Code Frequency** | 65.0 | 53.6 | 92.2 | 98.0 |
| **Acc. w/ Code** | 60.9 | 52.6 | 39.9 | 21.0 |
| **Acc. w/ Lang** | 35.0 | 17.2 | 61.5 | 40.0 |

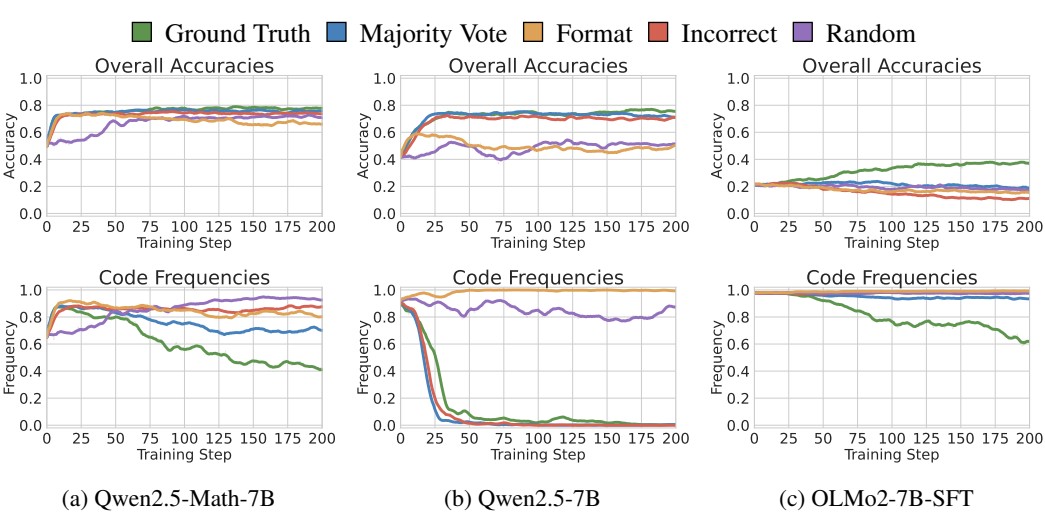

(a) Qwen2.5-Math-7B       (b) Qwen2.5-7B       (c) OLMo2-7B-SFT

Figure 5: We track the MATH-500 performance (left) and proportion of generated answers that contain Python code blocks (right) of a Qwen2.5-Math-7B model trained with various reward signals. As training with weak (format) or spurious rewards (random, incorrect) progresses, both accuracy and code frequency go up; RLVR with weak or spurious rewards primarily serves to upweight this pre-existing reasoning behavior. Training with ground truth also increases accuracy, but code frequency eventually decreases with continued training, suggesting a possible difference in improvement mechanism from weak or spurious rewards.

15 steps, correlated strongly with the accuracy improvements; random reward shows a more gradual increase in code frequency, but eventually reaches as high as 95.6%. We note that code reasoning frequency also sharply increases with RLVR training on ground truth labels, but gradually drops as the model's natural language reasoning accuracy increases, suggesting that the model is learning real knowledge from RLVR on high-quality, ground truth rewards. For the *Bad-Code* models, we find that *the reduction* of code reasoning frequency is highly correlated to performance gains. We provide additional details on the switch in reasoning strategies in Appendix H. These results further demonstrate that most of the improvements in RLVR with spurious rewards occur on the subset of questions where the model transitions from text-based to code-based reasoning after training.

## 4.3 INTERVENING EXPLICITLY ON CODE REASONING FREQUENCY

We have shown observationally that code reasoning frequency increases during RLVR, which correlates with increased test performance. Here, we investigate the *causal impact* of code reasoning by explicitly inducing it to occur more or less frequently. We present the impact of inducing additional code reasoning in the main paper, while the consequences of suppressing code reasoning are discussed in Appendix F. Experiments on both directions confirm our hypothesis.

**Inducing code reasoning significantly improves Qwen2.5-Math models' performance, and generally degrades other models.** We deliberately induce more frequent code reasoning behaviors via (1) prompting and (2) RLVR training. To induce with prompting, we force the models to begin their responses with "LET'S SOLVE THIS USING PYTHON." As shown in Table 2, the MATH-500

accuracy of Qwen2.5-Math-1.5B, Qwen2.5-Math-7B, and Qwen2.5-1.5B improved by 24.2%, 15.0%, and 10.0%, respectively. However, we see performance degradation for other models such as Llama and OLMo2. This drop is consistent with our earlier observation that the other models do not exhibit effective code reasoning behaviors (Section 4.1).

To induce increased coding reasoning frequency with RLVR, we train by assigning a positive reward if and only if a response contains the string "`python`", which we term as a *Python reward*. This intervention is effective; we find that Qwen2.5-Math-7B generated code reasoning solutions in > 99% of its answers after just 20 training steps. We show MATH-500 performance during training in Figure 6. Overall, we observe performance gains specifically in the Qwen2.5-Math models. Other models demonstrated limited improvement. For Qwen2.5-Math, we observe that inducing code reasoning with RLVR matches or exceeds the performance gains from inducing via prompting.

Table 2: Model performance on MATH-500 after augmenting the prompt to incentivize code reasoning. In this experiment, we force the model's first generated sentence to be "Let's solve this using Python." When applied to Qwen2.5-Math models, which have strong code reasoning priors, our "code-forcing" prompting strategy results in significantly increased test accuracy.

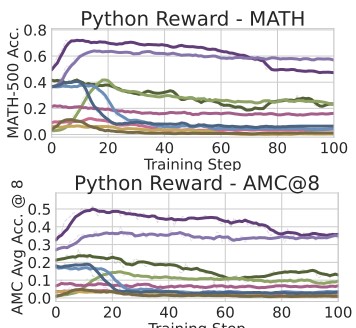

| Model | Original | Prompting | Abs. Diff. |
|---|---|---|---|
| Qwen2.5-Math-1.5B | 36.2% | 60.4% | +24.2% |
| Qwen2.5-Math-7B | 49.4% | 64.4% | +15.0% |
| Qwen2.5-1.5B | 3.0% | 13.0% | +10.0% |
| Qwen2.5-7B | 41.6% | 22.2% | −19.4% |
| Llama3.2-3B-Instruct | 36.8% | 8.2% | −28.6% |
| Llama3.1-8B-Instruct | 36.8% | 15.2% | −21.6% |
| OLMo2-7B | 9.0% | 7.8% | −1.2% |
| OLMo2-7B-SFT | 21.4% | 18.6% | −2.8% |

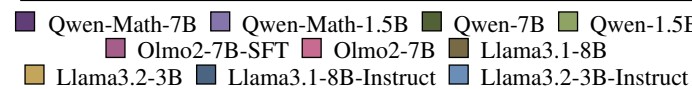

■ Qwen-Math-7B  ■ Qwen-Math-1.5B  ■ Qwen-7B  ■ Qwen-1.5B
■ Olmo2-7B-SFT  ■ Olmo2-7B  ■ Llama3.1-8B
■ Llama3.2-3B  ■ Llama3.1-8B-Instruct  ■ Llama3.2-3B-Instruct

Figure 6: Performance when using our Python reward to explicitly encourage the model to perform code reasoning. This improves performance in Qwen2.5-Math but not other models. Qwen2.5-Math-7B starts to generate > 99% code reasoning in 20 training steps.

## 5   THE CURIOUS CASE: INCORRECT AND RANDOM REWARDS

We present our hypotheses on how spurious rewards produce meaningful training signals in RLVR.

**Incorrect Rewards.**   We hypothesize two mechanisms that allow incorrect rewards to provide effective training signals. First, many incorrect labels remain close to ground truth values, providing positive reinforcement for largely correct reasoning. Second, incorrect labels partially function as format rewards: the model only receives positive rewards when its final answer is extractable and matches the incorrect labels.

**Random Rewards.**   First, we show in Appendix B that our random reward observations hold across different probabilities, confirming that it is a stable phenomenon. We then show the source of training signals under random rewards possibly originates from a bias in the training objective—**specifically, GRPO clipping bias can induce random reward training signals.** GRPO computes a normalized group-relative advantage over the rewards for gradient updates. The expectation of the advantage, which decides the direction and magnitude of the gradient updates during training, is zero (Appendix B.1.1). Even so, the expected gradient in the GRPO loss is *nonzero* due to the clipping mechanism in the loss (details in Appendix B.1.2). Following Yu et al. (2025), which finds that clipping bias reduces exploration and increases exploitation in RLVR with ground-truth labels, we hypothesize that clipping bias similarly reinforces high-priority behaviors related to the input domains (i.e., mathematics in our setup) under random rewards. We confirm this by showing that clipping bias increases the policy model's token probabilities for training answers, demonstrating enhanced reliance on prior knowledge (Appendix B.1.3).

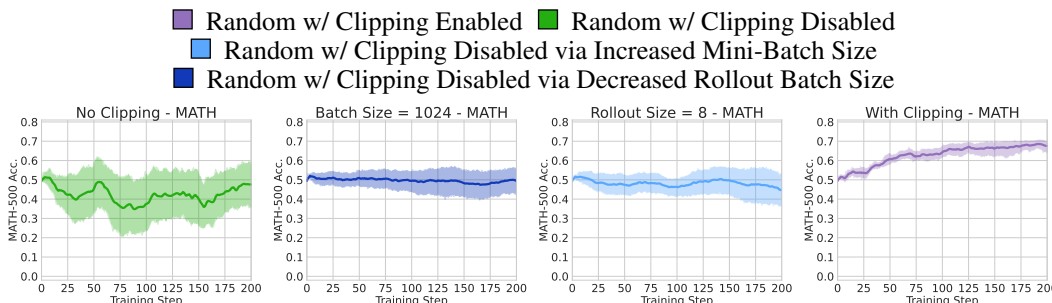

Figure 7: Average results over multiple random seeds when ablating the clipping term in GRPO with random rewards for Qwen2.5-Math-7B. We remove the clipping term through 3 methods: (i) disabling clipping in the implementation, (ii) increasing mini-batch size to match rollout size and (iii) reducing rollout size. Both (ii) and (iii) ensures that there is only 1 gradient update per rollout phase, so $\pi_\theta = \pi_{\text{old}}$, directly avoiding any clipping. From our observation, training with random rewards with clipping produces improved performance. Other runs without clipping fail to replicate the same behavior. In Appendix B.2, we show that no-clipping runs with different random seeds are highly stochastic, which can randomly achieve performance improvement or decrease by chance.

To validate the effect of clipping in GRPO, we conduct an ablation study by removing it during training. We achieve this by either (a) directly removing the clipping term in the loss calculation or (b) adjusting training and rollout batch sizes to ensure $\pi_\theta = \pi_{\text{old}}$, thus preventing clipping constraints from being triggered. As shown in Figure 7, under standard GRPO with clipping (purple line), random rewards increase code reasoning behavior under standard clipping, effectively concentrating the model on its existing reasoning pattern distribution. This concentration effect vanishes when clipping is disabled across different experimental conditions (other lines). This increase in code reasoning correlates with improved performance. Random rewards produce a $\sim 21\%$ performance gain. However, when clipping effects are eliminated, random rewards yield no robust improvement.

We show that GRPO's clipping mechanism can provide a meaningful training signal even from purely noisy rewards. We conjecture that the apparent training signal from random reward is an artifact of the optimization algorithm's bias toward exploiting existing priors learned during pretraining.

## 6   RELATED WORK

**Unsupervised Reinforcement Learning.**   Several approaches have explored unsupervised reinforcement learning. Prasad et al. (2024) introduced Self-Consistency Preference Optimization (ScPO), which trains models to prefer consistent answers over inconsistent ones on unsupervised problems. Similarly, Test-Time Reinforcement Learning (TTRL) (Zuo et al., 2025) leverages majority voting across sampled outputs to estimate pseudo-rewards, demonstrating significant performance improvements on mathematical reasoning tasks. EMPO (Zhang et al., 2025) adapts PPO (Schulman et al., 2017) or GRPO (Shao et al., 2024) for unsupervised RLVR by calculating the rewards based on minimizing the entropy of queries in the semantic space. These approaches suggest that internal model consistency can serve as an effective proxy for correctness, though they do not systematically investigate how different kinds of training rewards affect various model families during RLVR. We discuss broader applications of reinforcement learning to language models in Appendix K.

## 7   DISCUSSION

Our findings have three main implications: base model pretraining significantly affects RLVR outcomes; even corrupted or spurious supervision can enhance reasoning when it triggers useful existing behaviors; and effects observed in one model family may not generalize to others. Our work highlights the importance of (1) testing across multiple models with differing pretraining distributions, and (2) testing across multiple different baselines, such as format and random rewards, when evaluating reinforcement learning techniques.

## REPRODUCIBILITY STATEMENT

We open-source our code at https://anonymous.4open.science/r/Spurious_Rewards-9401. To ensure reproducibility, we will also release the training data, model checkpoints, and training logs.

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

APPENDIX

# A EXPERIMENTAL SETUP

## A.1 PRELIMINARIES ON GRPO

Denote the input prompt as $x$ and the corresponding model rollouts as $y$. Denote the $t$-th token in the $i$-th rollout $y$ as $y_t$, where $1 \le t \le |y|$. GRPO loss, as introduced by (Shao et al., 2024), contains the following components: For each prompt $x$, we compute the group-wise mean $\bar{r}_x$ and standard deviation $\sigma_x$. The normalized *group-relative advantage* is $\hat{A}(x, y) = \frac{r(x,y) - \bar{r}_x}{\sigma_x}$. Let $\pi_{\text{old}}$ be the behavior policy that produced the trajectories and $\pi_{\text{ref}}$ be a frozen reference policy (for example, the initial supervised model). We denote the token-level importance ratio by $\rho_t(y; \theta) = \frac{\pi_\theta(y_t|x, y_{<t})}{\pi_{\text{old}}(y_t|x, y_{<t})}$, where $t$ is the token index. With PPO-style clipping threshold $\epsilon_c$, KL-penalty weight $\lambda$, the surrogate objective maximized is

$$J(\theta) = \mathbb{E}_{x \sim \mathcal{D}, y \sim \pi_{\text{old}}(\cdot|x)} \left[ \sum_{t=1}^{|y|} \min \Big( \rho_t(y; \theta) \, \hat{A}(x, y), \text{clip}\big(\rho_t(y; \theta), 1 - \epsilon_c, 1 + \epsilon_c\big) \, \hat{A}(x, y) \Big) \right]$$
$$- \lambda \, \mathbb{E}_{x \sim \mathcal{D}} \Big[ \text{KL}\big(\pi_\theta \| \pi_{\text{ref}}\big) \Big]. \tag{1}$$

The KL regularization is typically adopted to ensure our model does not deviate too far from the frozen reference model. However, since we focus primarily on verifiable rewards and investigate the signals produced by these rewards, we assume that distribution shift is of lesser concern following (Liu et al., 2025). Moreover, KL regularization adds confounding factors to our analysis. Recent work has also shown that removing the KL term leads to better performance (Hu et al., 2025b). Therefore, we set $\lambda = 0$ in our work.

## A.2 DATASETS AND MODELS

We conduct our experiments on three canonical mathematical reasoning benchmarks:

- **MATH-500** (Hendrycks et al., 2021; Lightman et al., 2023): A standardized subset of the MATH dataset focusing on advanced mathematical reasoning, including problems from algebra, calculus, and number theory.
- **AMC** (Li et al., 2024): The American Mathematics Competition dataset contains diverse mathematical problems ranging from algebra to combinatorics and geometry. For this benchmark, we report model's average performance over 8 trials (avg@8).
- **AIME** (Li et al., 2024): The American Invitational Mathematics Examination dataset consists of challenging high-school level mathematical problems requiring multi-step reasoning. We include AIME questions from 2024 and 2025 for evaluation. For this benchmark, we report model's average performance over 8 trials (avg@8).

For our initial experiments, we primarily utilize the Qwen2.5-Math-7B model, a 7 billion parameter language model specifically tuned for mathematical reasoning tasks. We select this model (1) for its strong baseline performance on mathematical problems while remaining computationally efficient for multiple experimental iterations and (2) because it is frequently used in prior work (Zuo et al., 2025; Wang et al., 2025b).

## A.3 TRAINING CONFIGURATION

Unless otherwise specified, we train each model on 8 GPUs with a constant learning rate of 5e-7, a mini batch size (number of rollouts seen before a gradient update) of 128, and a rollout batch size (number of prompts we rollout at the same time) of 64. For each prompt, we collect 16 rollouts to compute advantages for GRPO update. We use a sampling temperature $\tau = 1$. We do not apply KL divergence loss or entropy loss in our training.

## A.4 DECODING CONFIGURATION

We use temperature of 0.0 for pass at 1, and temperature of 0.6 for pass at k during decoding time. All other hyperparameters are the default value following the Zuo et al. (2025) codebase.

## A.5 COMPUTATION RESOURCE

Each RLVR run in our experiment takes approximately 24 hours on 8 A100s.

## A.6 VISUALIZATION

In all plots, we smooth the metric of interest across ten training steps, and overlay the raw curves as transparent dashed lines. We do not smooth the step 0 performance, which corresponds to the performance of the base model before training. We report the smoothed value at the final training step as the final performance.

## A.7 EXPERIMENTAL SETUPS FOR TTRL AND ONE-SHOT RL

**TTRL setup.** We follow all hyperparameters from the original TTRL paper (Zuo et al., 2025), and train all models using their publicly released implementation. Due to compute constraints, we do not extensively sweep hyperparameters on the new base models we consider. Note that the TTRL accuracies are not directly comparable to our majority-vote reward results, because (a) TTRL trains on (unlabeled) test prompts while we train on a much larger set of distinct train prompts, and (b) TTRL updates labels during training based on the online policy's majority vote, while we assign labels once offline.

**Differences in our One-Shot RL setup from Wang et al. (2025b).** We note that for the one-shot RL settings in our paper, we do not apply entropy loss for more consistent setups with other experiments and better stability. However, Wang et al. (2025b) shows that the entropy loss may further improve the performance of one-shot RL and post-saturation generalization. We use the same training example $\pi_1$ (defined in Wang et al. (2025b)) for one-shot RLVR experiments on all models as in Wang et al. (2025b). Their work discusses that different models should possibly select different examples for more performant one-shot RLVR training, but use $\pi_1$ in all experiments due to high cost of trying different possible 1-shot examples. In particular, we note that this setup difference may cause a noticeable performance difference for Qwen2.5-7B model compared to the entropy loss setting used in Wang et al. (2025b) (71.2% with entropy loss vs. 54.8% without).

# B THE CURIOUS CASE OF RANDOM REWARDS

In this section, we study a special case of our spurious rewards, random rewards, where the reward is randomly assigned independently of the model rollouts. We first show that random rewards work across several non-zero reward probabilities (Figure 8), confirming that this is a stable observation. Next, we provide detailed gradient derivation in this special case and discuss our hypothesis on one of the potential sources of training signals with random rewards—the clipping bias in GRPO update.

**Random rewards with varying probabilities consistently improve performance.** We train Qwen2.5-Math-7B using GRPO with random rewards assigned by `Bernoulli`$(\gamma)$ variables, where $\gamma \in \{0.7, 0.5, 0.3, 0.001, 0\}$. Each response receives reward 1 with probability $\gamma$ and 0 otherwise. We find that all non-zero probability configurations successfully lead to significant performance gains after an initial period of exploration, yielding comparable performance to ground truth rewards (Figure 8). $\gamma = 0$ yields no improvement as expected, since constant rewards provide no learning signal. The convergence speed varies with $\gamma$, but all configurations eventually reach similar high-performance regimes, with accuracy improvements of 15-20 percentage points on MATH-500.

**Recap of notations.** Recall that we denote the input prompt as $x$ and the corresponding model rollouts as $\{y^{(1)}, \cdots, y^{(G)}\}$, where $G \in \mathbb{N}^+$ is the rollout size. In addition, we denote the $t$-th token in the $i$-th rollout $y^{(i)}$ as $y_t^{(i)}$, where $t \in \mathbb{N}^+$ and $1 \leq t \leq |y^{(i)}|$. The normalized group-relative advantage in GRPO is $\hat{A}(x, y) = \frac{r(x,y) - \bar{r}_x}{\sigma_x}$, where $\bar{r}_x$ is the group-wise mean and $\sigma_x$ is the group-

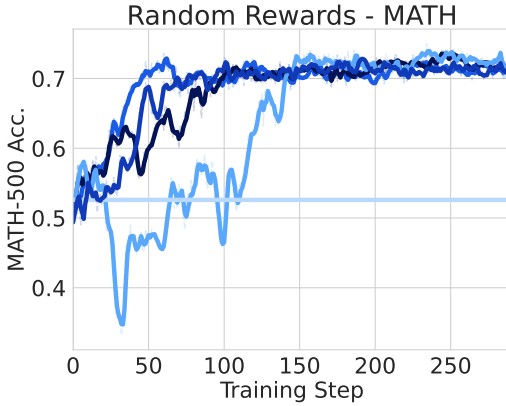

Figure 8: We train Qwen2.5-Math-7B using GRPO with random rewards of different probability $\gamma \in \{0.7, 0.5, 0.3, 0.001, 0\}$. We note that with the exception of $\gamma = 0$, which leads the model to not perform any learning, all other probability configuration successfully leads the model to achieve a significant performance gain after some period of random walk. Furthermore, once in the high-accuracy regime, the training procedure maintain model's performance.

■ Random $\gamma = 0.7$    ■ Random $\gamma = 0.5$
■ Random $\gamma = 0.3$    ■ Random $\gamma = 0.001$
□ Random $\gamma = 0$

wise standard deviation. The token-level importance ratio[1] is $\rho_t(y; \theta) = \frac{\pi_{\theta,x}(y_t)}{\pi_{\text{old},x}(y_t)}$, where $t$ is the token index.

### B.1 CONJECTURE: CLIPPING BIAS BRINGS TRAINING SIGNALS UNDER RANDOM REWARDS

#### B.1.1 EXPECTED ADVANTAGE OF RANDOM REWARDS

We investigate the expected advantage when the reward signal is assigned by a random $\texttt{Bernoulli}(\gamma)$ variable. For a prompt $x$ and rollouts $\{y^{(i)}\}_{i=1}^G$, the reward of each $y^{(i)}$ is $r(x, y^{(i)}) \sim \texttt{Bernoulli}(\gamma)$, so the expected average reward of $x$ is $\gamma$. Assuming $\hat{A}_i = 0$ when $\sigma_x = 0$, the sum of the normalized advantages over $G$ rollouts $\sum_{i=1}^G \hat{A}(x, y^{(i)}) := \sum_{i=1}^G \frac{r(x,y^{(i)}) - \bar{r}_x}{\sigma_x} = \frac{\sum_{i=1}^G r(x,y^{(i)}) - G \cdot \bar{r}_x}{\sigma_x} = 0$ by construction. Furthermore, for i.i.d. rewards that are independent of the provided samples, such as the random rewards used in our experiments, $\mathbb{E}(\sum_{i=1}^G \hat{A}(x, y^{(i)})) = G \cdot \mathbb{E}(\hat{A}) = 0 \implies \mathbb{E}(\hat{A}) = 0$.

The clipping term in GRPO loss (Equation 1) prevents excessive deviation from the previous policy, stabilizing training. Recent work suggests this term, which operates on the ratio $\rho_t(y; \theta) = \pi_{\theta,x}(y_t)/\pi_{\text{old},x}(y_t)$, introduces bias toward exploitation in the case of ground truth reward (Yu et al., 2025). Here, we analyze the bias in the settings of the random rewards, where each rollout receives an advantage that is independent of the rollout.

We note that the loss is no longer differentiable everywhere with clipping. In practical implementations, e.g., PyTorch, the gradients will be automatically set to 0 when the value is clipped. For simplicity, we discuss the gradient of the loss function by assuming 0 gradient at the non-differentiable points.

#### B.1.2 GRADIENT DERIVATION OF CLIPPING BIAS UNDER RANDOM REWARDS

In this section, we derive the training signals induced by the clipping factor, which we name as a "clipping bias". Specifically, we define the clipping bias as the difference in the expected gradients after adding clipping to the GRPO objective[2]:

$$\texttt{Bias}(\nabla_\theta J(\theta)) = \mathbb{E}_{x,y}[\nabla_\theta J(\theta)] - \mathbb{E}_{x,y}[\nabla_\theta L^{\text{unclipped}}(\theta)].$$

As we derived above, $\mathbb{E}(\hat{A}(x, y_j)) = 0$. Assuming a simple loss with no clipping, the expectation of the policy gradient without the clipping term is also trivially zero given that $\hat{A}$ is independent of other

---

[1]For simplicity, we denote $\pi(y_t | x, y_{<t})$ as $\pi_x(y_t)$, and similarly $\pi_\theta(y_t | x, y_{<t})$ as $\pi_{\theta,x}(y_t)$, $\pi_{\text{old}}(y_t | x, y_{<t})$ as $\pi_{\text{old},x}(y_t)$.

[2]For simplicity, we denote $\mathbb{E}_{x \sim \mathcal{D}, y \sim \pi_{\text{old}}(\cdot | x)}$ as $\mathbb{E}_{x,y}$

variables:

$$\mathbb{E}_{x,y}[\nabla_\theta L^{\text{unclipped}}(\theta)]$$

$$=\mathbb{E}_{x,y}\left[\frac{1}{|y|}\sum_{t=1}^{|y|}\nabla_\theta \rho_t(y;\theta)\,\hat{A}(x,y)\right]$$

$$=\mathbb{E}(\hat{A})\mathbb{E}_{x,y}\left[\frac{1}{|y|}\sum_{t=1}^{|y|}\nabla_\theta \rho_t(y;\theta)\right]$$

$$=0.$$

Therefore, the clipping bias under random reward is

$$\text{Bias}(\nabla_\theta J(\theta)) = \mathbb{E}[\nabla_\theta J(\theta)] - \mathbb{E}[\nabla_\theta L^{\text{unclipped}}(\theta)] = \mathbb{E}[\nabla_\theta J(\theta)].$$

Next, we compute the exact form of this clipping bias, i.e., the gradient in GRPO with random rewards. Recall from Eq. 1, the surrogate objective that we aim to maximize is:

$$J(\theta) = \min\Big(\rho_t(y;\theta)\,\hat{A}(x,y), \text{clip}\big(\rho_t(y;\theta), 1-\epsilon_c, 1+\epsilon_c\big)\,\hat{A}(x,y)\Big).$$

For simplicity, we further denote $R_\theta = \rho_t(y;\theta) = \frac{\pi_{\theta,x}(y_t)}{\pi_{\text{old},x}(y_t)}$ to be the token-level importance ratio, $C = \text{clip}(R_\theta, 1-\epsilon_c, 1+\epsilon_c)$ to be the clipping term, and $\hat{A}(x,y)$ as $\hat{A}$. So we have $J(\theta) = \min(R_\theta \cdot \hat{A}, C \cdot \hat{A})$. We now analyze the gradient of GRPO loss with respect to the policy model parameters $\theta$ based on the sign of $\hat{A}$.

As discussed in Section B.1, the loss function is not differentiable everywhere. Therefore, we discuss the gradients with respect to the differentiable regions below and set the gradients to 0 at the non-differentiable points at the end.

**Case 1:** $\hat{A} \geq 0$. When $\hat{A} \geq 0$, we can directly take $\hat{A}$ out of the $\min$ function. Thus,

$$J(\theta) = \min(R_\theta \cdot \hat{A}, C \cdot \hat{A}) = \hat{A} \cdot \min(R_\theta, C).$$

Since $C$ is defined as $C = \text{clip}(R_\theta, 1-\epsilon_c, 1+\epsilon_c)$, we have:

- If $R_\theta < 1 - \epsilon_c$, then $C = 1 - \epsilon_c$. Thus, $\min(R_\theta, C) = R_\theta$.
- If $1 - \epsilon_c \leq R_\theta \leq 1 + \epsilon_c$, then $C = R_\theta$. Thus, $\min(R_\theta, C) = R_\theta$.
- If $R_\theta > 1 + \epsilon_c$, then $C = 1 + \epsilon_c$. Thus, $\min(R_\theta, C) = 1 + \epsilon_c$.

Combining these, when $\hat{A} \geq 0$, the objective is:

$$L^+(\theta) = \hat{A} \cdot \begin{cases} R_\theta, & \text{if } R_\theta < 1 + \epsilon_c, \\ 1 + \epsilon_c, & \text{if } R_\theta > 1 + \epsilon_c. \end{cases}$$

The gradient with respect to $\theta$ for this case is:

$$\nabla_\theta L^+(\theta) = \hat{A} \cdot \begin{cases} \nabla_\theta R_\theta, & \text{if } R_\theta < 1 + \epsilon_c, \\ 0, & \text{if } R_\theta > 1 + \epsilon_c. \end{cases}$$

**Case 2:** $\hat{A} < 0$. When $\hat{A} < 0$, multiplying by $\hat{A}$ flips the $\min$ function. So,

$$L^-(\theta) = \min(R_\theta \cdot \hat{A}, C \cdot \hat{A}) = \hat{A} \cdot \max(R_\theta, C).$$

Applying a similar analysis, when $\hat{A} < 0$, the gradient with respect to $\theta$ for this case is:

$$\nabla_\theta L^-(\theta) = \hat{A} \cdot \begin{cases} 0, & \text{if } R_\theta < 1 - \epsilon_c, \\ \nabla_\theta R_\theta, & \text{if } R_\theta > 1 - \epsilon_c. \end{cases}$$

**Combining the cases.** Together, the gradient is:

$$\nabla_\theta J(\theta) = \hat{A} \cdot \begin{cases} \nabla_\theta R_\theta, & \text{if } \hat{A} \geq 0 \text{ and } R_\theta < 1 + \epsilon_c, \\ 0, & \text{if } \hat{A} \geq 0 \text{ and } R_\theta > 1 + \epsilon_c, \\ 0, & \text{if } \hat{A} < 0 \text{ and } R_\theta < 1 - \epsilon_c, \\ \nabla_\theta R_\theta, & \text{if } \hat{A} < 0 \text{ and } R_\theta > 1 - \epsilon_c. \end{cases}$$

Then, the clipping bias, $\mathtt{Bias}(\nabla_\theta J(\theta))$, is

$\mathtt{Bias}(\nabla_\theta J(\theta))$

$= \mathbb{E}_{\hat{A},x,y}[\nabla_\theta J(\theta)]$

$= P(\hat{A} \geq 0) \cdot \mathbb{E}_{\hat{A}>0,x,y}[\nabla_\theta L^+(\theta)] + P(\hat{A} < 0) \cdot \mathbb{E}_{\hat{A}<0,x,y}[\nabla_\theta L^-(\theta)] + 0$

$= \mathbb{E}_{x,y}\left[\begin{cases} P(\hat{A} \geq 0) \cdot \mathbb{E}_{\hat{A}\geq0}[\hat{A}] \cdot \nabla_\theta R_\theta, & \text{if } R_\theta < 1 + \epsilon_c, \\ 0, & \text{if } R_\theta > 1 + \epsilon_c. \end{cases}\right]$

$\quad + \mathbb{E}_{x,y}\left[\begin{cases} 0, & \text{if } R_\theta < 1 - \epsilon_c, \\ P(\hat{A} < 0) \cdot \mathbb{E}_{\hat{A}<0}[\hat{A}] \cdot \nabla_\theta R_\theta, & \text{if } R_\theta > 1 - \epsilon_c. \end{cases}\right]$

$= \mathbb{E}_{x,y}\left[\begin{cases} P(\hat{A} \geq 0) \cdot \mathbb{E}_{\hat{A}\geq0}[\hat{A}] \cdot \nabla_\theta R_\theta, & \text{if } R_\theta < 1 - \epsilon_c, \\ (P(\hat{A} \geq 0) \cdot \mathbb{E}_{\hat{A}\geq0}[\hat{A}] + P(\hat{A} < 0) \cdot \mathbb{E}_{\hat{A}<0}[\hat{A}]) \cdot \nabla_\theta R_\theta, & \text{if } 1 - \epsilon_c < R_\theta < 1 + \epsilon_c, \\ P(\hat{A} < 0) \cdot \mathbb{E}_{\hat{A}<0}[\hat{A}] \cdot \nabla_\theta R_\theta, & \text{if } R_\theta > 1 + \epsilon_c. \end{cases}\right]$

By definition, $\hat{A}$ is a normalized distribution and $\mathbb{E}(\hat{A}) = 0$ from Appendix B.1.1, so

$$\mathbb{E}(\hat{A}) = P(\hat{A} \geq 0)\, \mathbb{E}_{\hat{A}\geq0}(\hat{A}) + P(\hat{A} < 0)\, \mathbb{E}_{\hat{A}<0}(\hat{A}) = 0, \text{ and}$$

$$P(\hat{A} \geq 0)\, \mathbb{E}_{\hat{A}\geq0}(\hat{A}) = -P(\hat{A} < 0)\, \mathbb{E}_{\hat{A}<0}(\hat{A}) \geq 0.$$

We denote $\mu = P(\hat{A} \geq 0)\, \mathbb{E}_{\hat{A}\geq0}(\hat{A}) = -P(\hat{A} < 0)\, \mathbb{E}_{\hat{A}<0}(\hat{A})$, which by definition is positive. And we substitute $R_\theta = \frac{\pi_{\theta,x}(y_t)}{\pi_{\text{old},x}(y_t)}$ in the conditions. Finally, we set the gradients on non-differentiable points to 0 and have

$$\mathtt{Bias}(\nabla_\theta J(\theta)) = \mu \cdot \mathbb{E}_{x,y}\left[\begin{cases} \nabla_\theta R_\theta, & \text{if } \pi_{\theta,x}(y_t) < \pi_{\text{old},x}(y_t) \cdot (1 - \epsilon_c), \\ 0, & \text{if } \pi_{\text{old},x}(y_t) \cdot (1 - \epsilon_c) \leq \pi_{\theta,x}(y_t) \\ & \qquad \leq \pi_{\text{old},x}(y_t) \cdot (1 + \epsilon_c), \\ -\nabla_\theta R_\theta, & \text{if } \pi_{\theta,x}(y_t) > \pi_{\text{old},x}(y_t) \cdot (1 + \epsilon_c). \end{cases}\right]$$

Recall that $R_\theta = \rho_t(y; \theta) = \frac{\pi_{\theta,x}(y_t)}{\pi_{\text{old},x}(y_t)}$ and both $\pi_{\theta,x}(y_t)$ and $\pi_{\text{old},x}(y_t)$ in range $[0, 1]$. Therefore, we observe that there is a *positive gradient* $\big($gradient that increases $\pi_{\theta,x}(y_t)\big)$ if $R_\theta < 1 - \epsilon_c$, and a *negative gradient* $\big($gradient that decreases $\pi_{\theta,x}(y_t)\big)$ if $R_\theta > 1 + \epsilon_c$, which means the clipping bias discourages the model from leaving the clipping region. In the rest of the paper, for simplicity we refer to the positive gradient case as "positive gradient bias" and the other way as "negative gradient bias".

### B.1.3 Clipping Creates Asymmetric Updates Towards Model Prior Knowledge

In this section, we provide further evidence for our hypothesis that the clipping bias increases the likelihood of high-probability rollouts, similar to Yu et al. (2025)'s analysis on GRPO with ground-truth labels. We showcase this trend with a simple example. Consider a case where a token has a high-probability $\pi_{\text{old},x}(y_t) = 0.85$ and an $\epsilon_c = 0.2$ that we adopt in our experiments. Then, the upper threshold from the bias formula becomes $\pi_{\text{old},x}(y_t) \cdot (1 + \epsilon_c) = 1.02$. Since the probability output by the policy model cannot exceed 1, the upper clipping threshold of 1.02 is never reached. Thus, the gradient bias is nonnegative for this token, leading to a net positive gradient bias on the policy model, which leads to increase in probability on this token.

On the other hand, for a low-probability token where $\pi_{\text{old},x}(y_t) = 0.02$, the policy model receives a negative gradient bias when $\pi_{\theta,x}(y_t) > \pi_{\text{old},x}(y_t) \cdot (1 + \epsilon_c) = 0.024$; and positive gradient bias

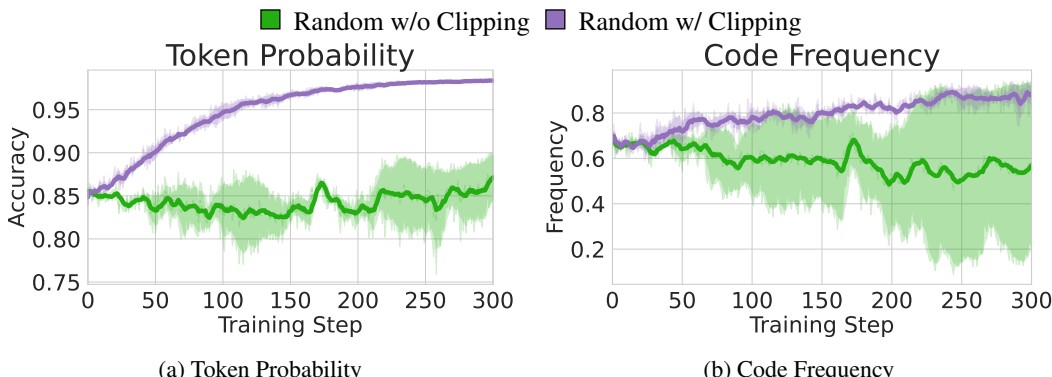

(a) Token Probability          (b) Code Frequency

Figure 9: Token probability and code frequency of random rewards with (purple) and without (green) clipping on Qwen2.5-Math-7B.

when $\pi_{\theta,x}(y_t) < \pi_{\text{old},x}(y_t) \cdot (1 - \epsilon_c) = 0.016$. The low threshold for negative gradient bias makes penalties more likely than in the high-probability case.

The clipping range width scales linearly with the original probability: higher-probability tokens have wider ranges and face fewer penalties. This asymmetric treatment prevents low-probability samples from receiving substantial upweighting during training, causing the model to concentrate probability mass on its existing distribution.

**Empirical validation.** We empirically validate our conjecture by disabling the clipping term in the GRPO loss, and observe differences in the training dynamics. Specifically, we focus on *Token Probability* $\pi_\theta$ and *Code Frequency* as defined below.

Recall that $\pi_{\text{old}}$ is the old policy from the previous training step and the $\pi_\theta$ the current policy, let $G$ be the number of rollouts generated per prompt; let $B$ be the mini batch size, which is the number of prompts on which a single gradient update is performed; and let $M$ be the number of gradient updates from the old policy, $\pi_{\text{old}}$, to the new policy model, $\pi_\theta$. Note that $G \cdot B \cdot M$ is the total number of rollouts generated by $\pi_{\text{old}}$. The average token probability of $\pi_\theta$ over all rollouts $\{y^{(k)}\}$ at any step can be expressed as:

$$\pi_{\theta,x}(y) = \frac{1}{M} \sum_{i=1}^{M} \frac{1}{B} \sum_{j=1}^{B} \frac{1}{G} \sum_{k=1}^{G} \pi_{\theta_i,x_j}(y^{(k)}).$$

Figure 9 (a) shows that the average token probability mass increases for the standard loss of GRPO with clipping (purple line), but stays relatively constant when clipping is disabled. In addition, we log the frequency of code reasoning throughout the RLVR training in Figure 9 (b). We hypothesize that the increase in average token probability correlates with the increase in code reasoning behavior, which, as we conjectured in Section 4, enhances the performance of the model, as we empirically show in Figure 5 of Section 4.2.

**Why only Qwen-Math models benefit.** The clipping bias mechanism operates on all models, but its effectiveness depends on what behaviors get amplified. The bias systematically favors a model's pre-existing high-probability behaviors, but only benefits performance if those behaviors correlate with correctness. For Qwen2.5-Math models, code reasoning occurs in 65% of responses and correlates strongly with correctness (64% accuracy vs. 29% without code). We observe that, when RLVR with clipping bias increases the model's code reasoning frequency to >90%, performance improves substantially. *No-Code models* (Llama, OLMo2-7B) generate no code reasoning, so clipping bias has no beneficial pattern to amplify. On the other hand, *Bad-Code models* (OLMo2-7B-SFT) generate code 98% of the time but with 21% accuracy vs. 40% for natural language, indicating that good performance does not rely on code reasoning. Overall, the clipping bias is universal, but its impact on performance depends entirely on whether the model's dominant pre-existing behaviors happen to be effective reasoning strategies.

**Implications.** Our findings reveal that GRPO's clipping mechanism provides a meaningful training signal even from pure noisy rewards, by systematically favoring the model's pre-existing behavioral patterns. This suggests that the apparent "reward signal" in random reward training is actually an artifact of the optimization algorithm's bias toward exploiting learned priors rather than exploring new behaviors. This mechanism explains why random rewards work for Qwen2.5-Math models (which have strong code reasoning priors) but fail for other model families lacking these pre-existing capabilities. The training algorithm amplifies whatever reasoning patterns already correlate with correctness, regardless of the reward signal's actual informativeness.

### B.2 HIGH STOCHASTICITY IN RLVR WITHOUT CLIPPING

In this section, we empirically show that training without clipping has stochastic training dynamics and can sometimes obtain performance improvement despite the expected gradient being zero. Following the experimental setup in Section 5, we disable the clipping effect through the following interventions: (i) removing clipping from the loss implementation, (ii) increasing mini-batch size to match rollout size, and (iii) decreasing the rollout size to match mini-batch size across multiple random seeds. We report the results in Figure 10 across different random seeds.

When clipping is disabled by adjusting batch size, model performance remains stable within a consistent range without converging toward either extreme of the accuracy spectrum. These runs involve 8 times fewer gradient updates due to the batch configuration.

In contrast, removing clipping from the implementation produces extreme stochasticity, occasionally resulting in high-performance convergence across multiple runs. While the exact mechanism remains unclear, we hypothesize that inherent randomness in RLVR training contributes to these observations. The group size $G = 16$ used in GRPO training creates high variance in gradient updates. Combined with the instability of unclipped training, this variance may accidentally reinforce certain reasoning patterns, leading to improved performance in some runs. We leave a systematic analysis of this behavior to future work.

## C ADDITIONAL RESULTS ON AMC

We supplement additional AMC results for non-Qwen models in Figure 11. The trends are consistent with the MATH-500 results that are shown in Section 3.

## D ADDITIONAL RESULTS ON AIME 2024 AND AIME 2025

We present additional results on the AIME benchmarks, which are challenging math Olympiad tests containing significantly harder problems than those in MATH-500 or AMC. We evaluate average@8 accuracy on AIME24 and AIME25 (Li et al., 2024). AIME25 was created after the release date of all models considered in our study. Thus, evaluating performance on AIME25 allows us to control the risk that our models' have seen similar problems during web pretraining. We evaluate the trained models from Section 2 and Section 3. We show results on Qwen2.5-Math models in Figure 12 and on the 8 additional models from Section 3 in Figure 13.

### D.1 SPURIOUS REWARDS YIELD SIGNIFICANT RLVR GAINS ON QWEN2.5-MATH

As shown in Figure 12, spurious rewards can consistently yield performance gains on Qwen-Math models on AIME24. Intriguingly, we find that any AIME24 gains achievable from training Qwen models with spurious rewards largely vanish when evaluating on AIME 2025. We speculate that AIME25 contains questions that are more out-of-distribution to Qwen's pretrained knowledge; spurious rewards—which largely serve to elicit existing knowledge—hence no longer provide benefit.

### D.2 (LACK OF) GENERALIZATION TO OTHER MODELS

Overall, results on other models are largely consistent with our earlier findings on AMC and MATH-500 (§3). Only Qwen2.5-7B, Qwen2.5-1.5B, and Llama3.1-8B-Instruct exhibit any notable gains from any reward signals on AIME. For Qwen2.5 models, weak and spurious rewards can yield

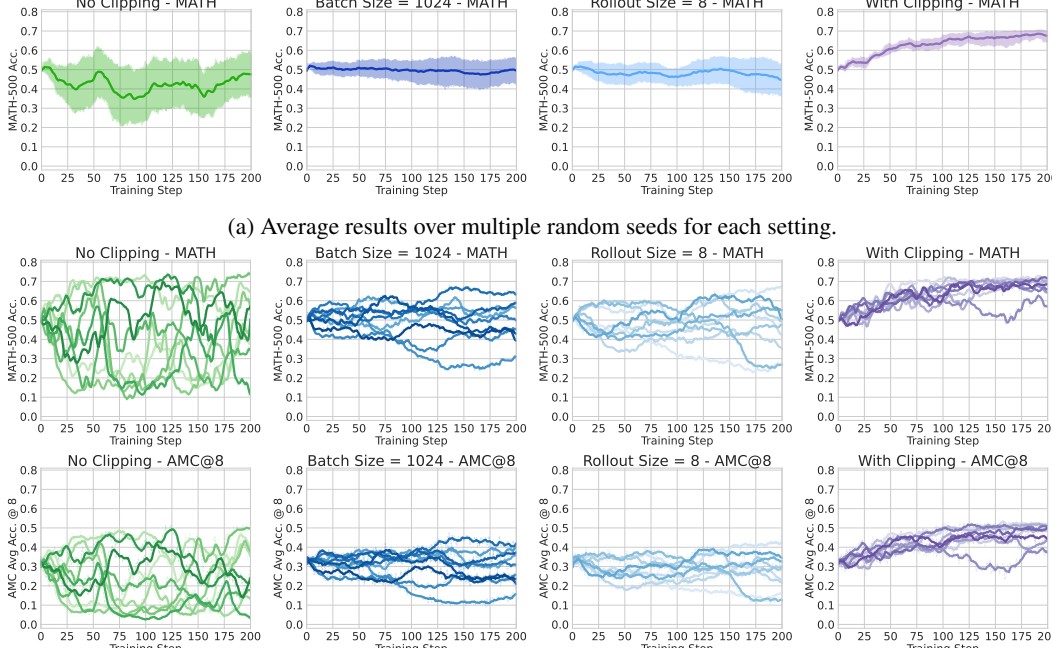

(a) Average results over multiple random seeds for each setting.

(b) Turning off clipping with 8 gradient updates per rollout. (c) Increasing batch size to ensure only 1 gradient update per rollout. (d) Decreasing rollout batch size to ensure only 1 gradient update per rollout. (e) Default GRPO loss with random reward.

Figure 10: RLVR performance on random rewards with disabled clipping across multiple different seeds. Models without clipping show no meaningful performance improvement on average, while models with clipping demonstrate consistent performance gains using random rewards. Disabling clipping in implementation produces high variance in results, occasionally yielding high accuracy scores. Experiments with adjusted batch sizes exhibit greater stability but involve 8 times fewer gradient updates than the disabled clipping experiments.

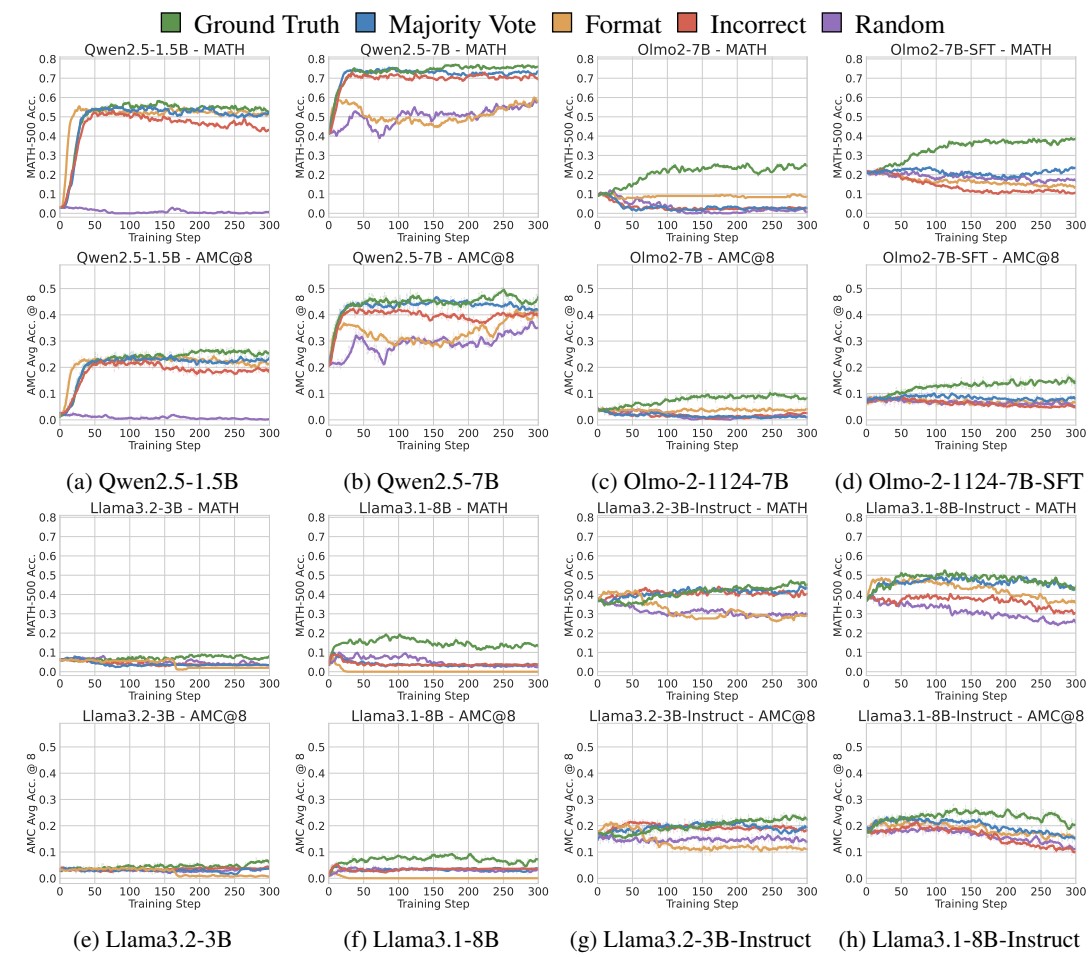

Figure 11: Varying rewards across additional model classes. Spurious rewards remain effective on general-purpose Qwen2.5 models but generally fail to yield any gains on other model families. The performance improvements on non-Qwen2.5 models are substantially smaller compared to those observed in the Qwen2.5 family.

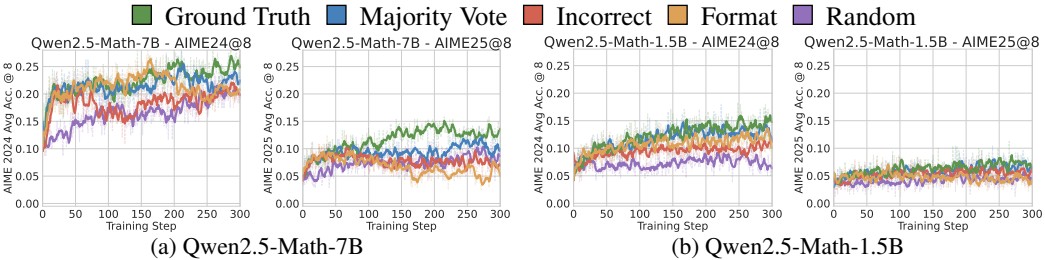

Figure 12: Qwen2.5-Math Model performance on AIME 2024 and AIME 2025.

gains; for example, format reward for Qwen2.5-1.5B and incorrect reward for Qwen2.5-7B. For Llama3.1-8B-Instruct, only standard rewards (e.g., ground truth and majority vote) yield gains. As observed above, performance and gains from RLVR training are lower across the board for all models on AIME25.

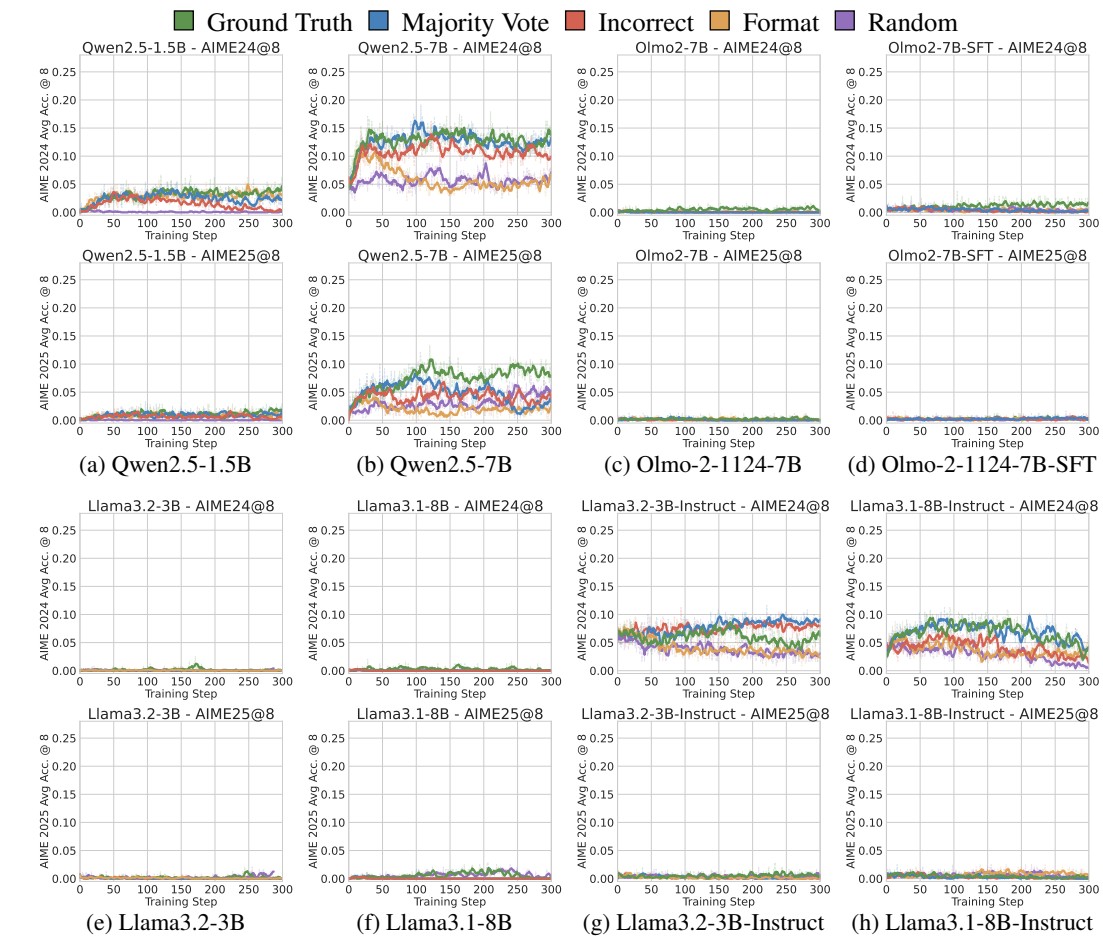

Figure 13: Varying rewards across additional model classes on the AIME 2024 and AIME 2025 benchmarks. Note that AIME 2025 was released after all the models' release dates, serving as a good resource to examine any dataset contamination phenomenon. Weak and spurious rewards (except for random) remain effective on general-purpose Qwen2.5 models but generally fail to yield any gains on other model families. The performance improvements on non-Qwen2.5 models are substantially smaller compared to those observed in the Qwen2.5 family. Note that the AIME benchmarks contain 30 questions each, so small differences in accuracy (less than $\sim$ 2 pp.) may not be significant.

## E  EXAMINATION OF EXISTING METHODS ON MODELS BEYOND QWEN

In this section, we examine the existing methods—TTRL and 1-shot RL—that apply weak supervision during RL, as shown in Figure 15. We find both methods are significantly effective on Qwen models, but not others.

## F  COMPOUND REWARDS THAT INHIBIT CODE REASONING

**Inhibiting code reasoning during RLVR with spurious rewards can reduce gains on Qwen2.5-Math-7B, but increase gains on other models.**   We hypothesized that code reasoning is part of the source of weak and spurious rewards gains. Contrapositively, *penalizing* code frequency could potentially reduce the gains of these rewards in Qwen2.5-Math. To test this, we designed compound rewards that intersect each weak or spurious reward with a *no Python reward*, so that a response is rewarded if and only if (1) it satisfies the original spurious reward condition and (2) it does not contain the string "python". Consistent with our hypothesis, format rewards cease to improve Qwen2.5-Math-7B when paired with the no code reward (Figure 16c). The spurious incorrect compound reward matches the original incorrect reward on MATH-500; on more difficult benchmarks such as AMC

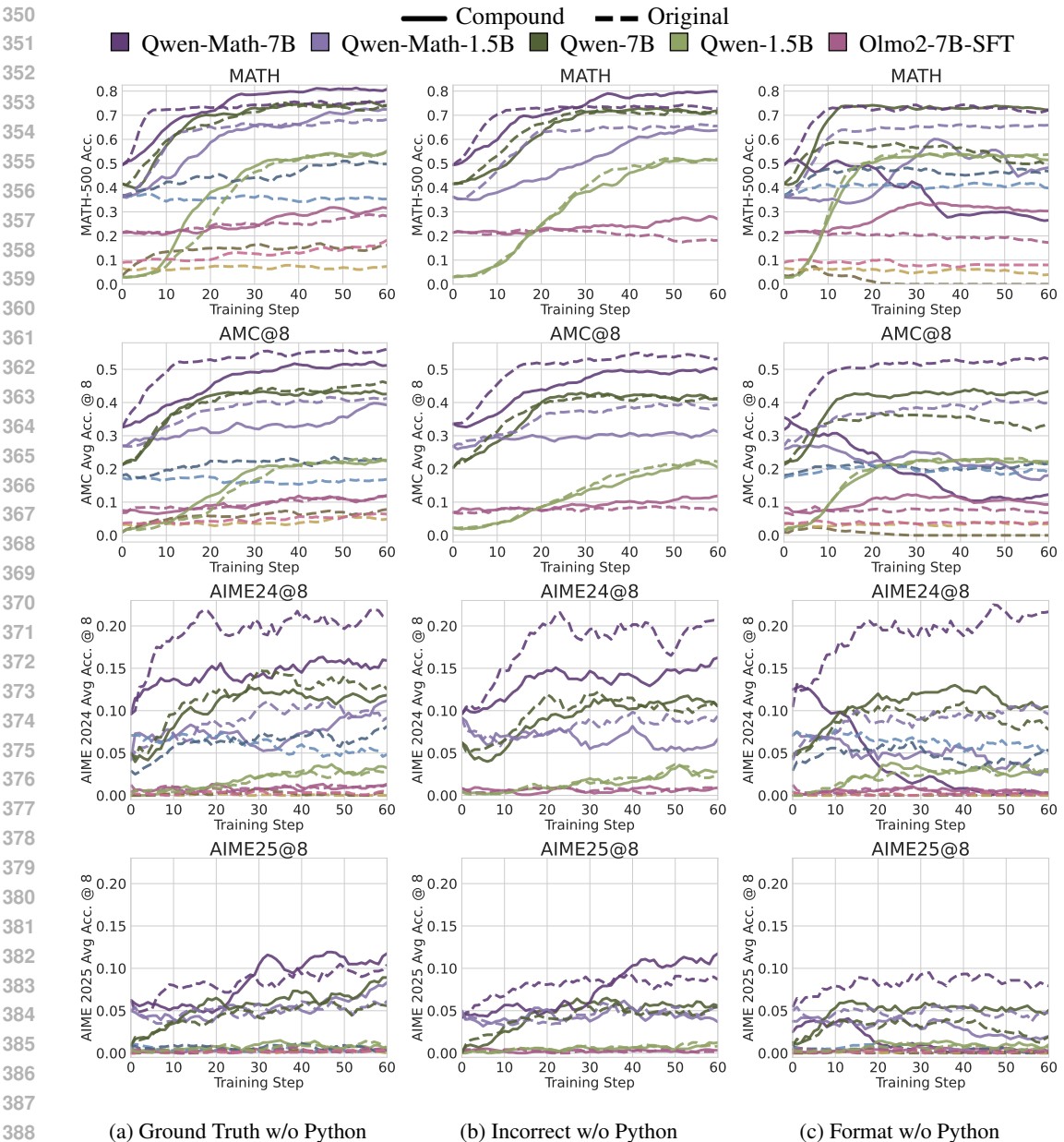

(a) Ground Truth w/o Python  (b) Incorrect w/o Python  (c) Format w/o Python

Figure 14: We present additional compound reward results on (1) smaller 1.5B Qwen models and (2) AMC and AIME benchmarks. See Section 4.3 for full details of the setup. Our compound rewards intersect (a) our original rewards with a (b) *no Python reward* that only rewards responses without Python code. Overall, our findings here are largely consistent with those from our main text. Note that AIME is a very small test set, so small differences in accuracy (less than $\sim 2$ percentage points) may not be meaningful.

and AIME, gains also persist but are reduced (Figures 16b, 14). Thus, while ablating code reasoning does hurt the incorrect reward as predicted, we posit that other beneficial behaviors (e.g., reduced repetition, see Appendix G) may still be superficially elicited. In addition, ground-truth rewards still yield gains, suggesting benefits beyond eliciting code reasoning (consistent with the code-frequency trends in Figure 5).

Intriguingly, for *Bad-Code* models Qwen2.5-7B and OLMo2-7B-SFT, compound rewards often *outperform* the originals. Most strikingly, OLMo2-7B-SFT—which degrades under standalone format or incorrect rewards—gains +8.9 and +5.5 points, respectively, once the no-code reward is added.

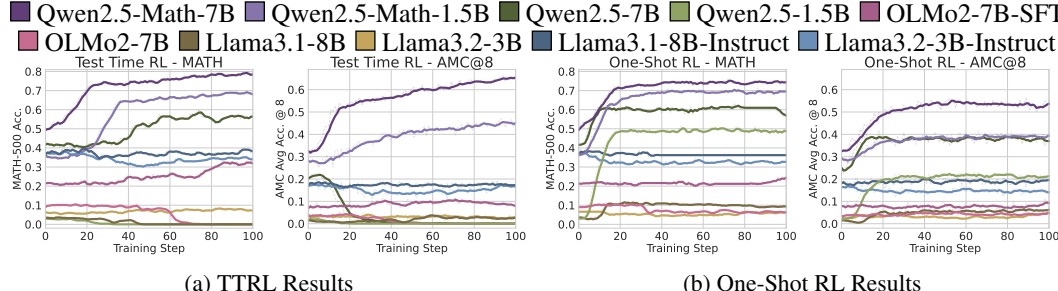

(a) TTRL Results      (b) One-Shot RL Results

Figure 15: We evaluate two recent weak supervision RL methods—TTRL (Zuo et al., 2025) and One-Shot RL (Wang et al., 2025b)—on diverse base models. We find that the proposed training rewards can consistently work on Qwen models. Yet with few exceptions, those same proposed signals often yield no gains on other model families, mirroring the limited generalization observed when training with our own spurious rewards. See Appendix A.7 for setup details.

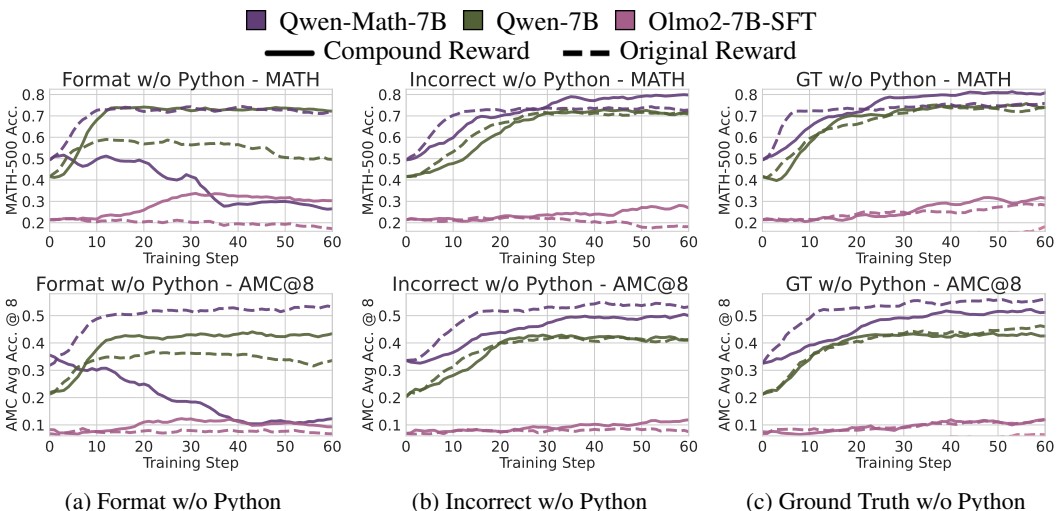

(a) Format w/o Python      (b) Incorrect w/o Python      (c) Ground Truth w/o Python

Figure 16: RLVR with compound rewards that intersect (i) our original rewards with a (ii) *no Python reward* that only rewards responses without Python code. We defer corroborating results on AIME and more models to Appendix F.

We hypothesize that this is because Qwen2.5-7B and OLMo2-7B-SFT exhibit weak code reasoning before RLVR training, so compound rewards explicitly downweight a behavior that is suboptimal for these models.

In Figure 14, we present additional results for our compound rewards on (1) more models (Qwen2.5-1.5B, Qwen2.5-Math-1.5B) and (2) more benchmarks (AIME24, AIME25). Overall, results corroborate our analysis in Section 4.3; Qwen2.5-Math-1.5B follows the same overall trends as Qwen2.5-Math-7B. Compound rewards also continue to benefit Qwen2.5-1.5B, where gains are comparable to the gains from using the original reward. This slightly contrasts our observations in Qwen2.5-7B, where compound rewards often yielded stronger gains. We conjecture that Qwen2.5-1.5B is stronger at code reasoning, which is consistent with our finding in Section 4.3 that inducing code reasoning via prompting improves performance in in Qwen2.5-1.5B but not Qwen2.5-7B. We are unsure of the exact reason for this discrepancy between these two models (which have the same pretraining data) and leave it for future work.

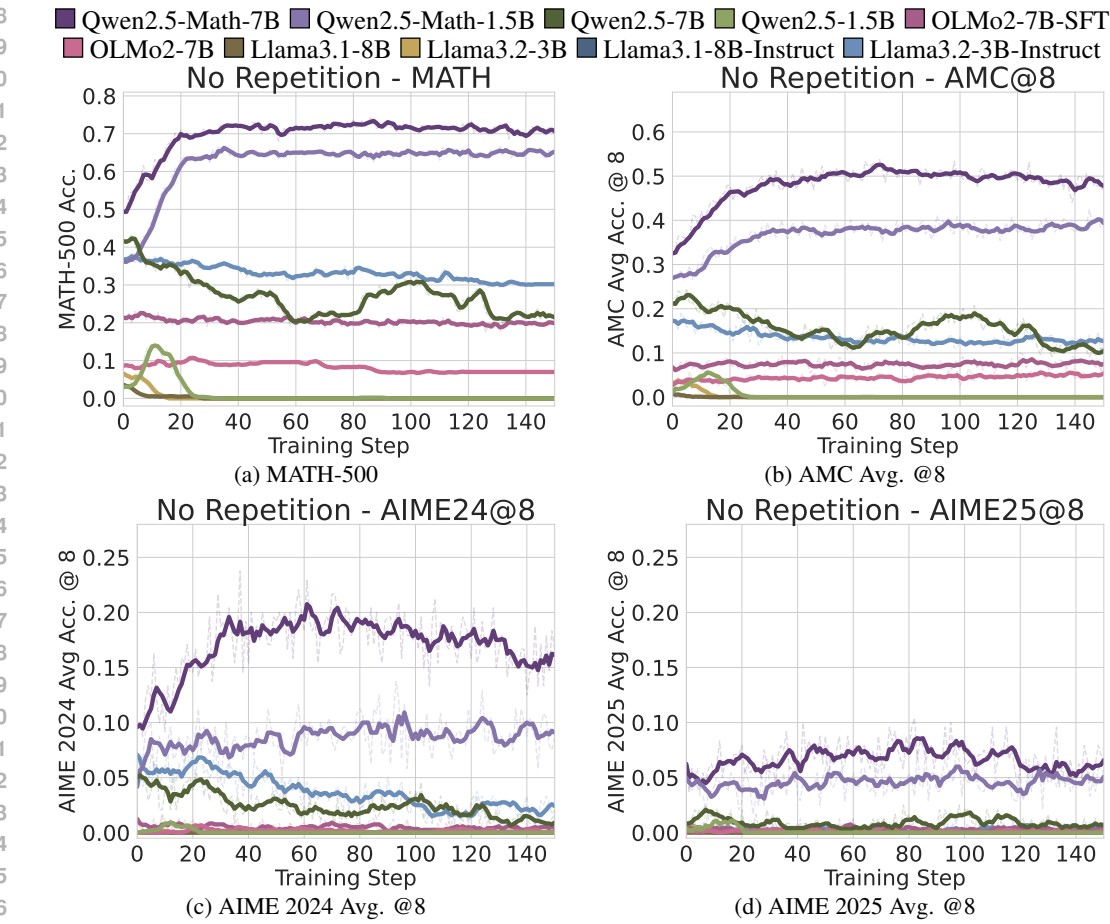

Figure 17: We design a new type of reward, *no-repetition reward*, which assign a score of 1 to responses that do not contain obvious repetition and 0 to responses that contain obvious string repetition. We find, no-repetition reward effectively improves the performance of Qwen2.5-Math, while not others.

## G  BEYOND CODE REASONING: ANOTHER BENEFICIAL PATTERN THAT RLVR CAN EASILY ELICIT

In the main paper, we show that RLVR with spurious rewards can improve Qwen2.5-Math's performance by surfacing useful reasoning patterns learned during pretraining, and we use code reasoning as a standout example. We note that code reasoning is one distinctive reasoning representation, but not the only one. In this section, we briefly discuss another pattern—no repetition—that can also be easily elicited by RLVR in addition to code reasoning.

We observe that Qwen2.5-Math models have a relatively higher tendency to produce repetitive outputs compared to Llama3 and Olmo2 models. We find that answers with code reasoning often do not have this issue. Therefore, we further study the effect of purely discouraging repetition in answers. To study this, we design a new *repetition reward* that returns a score of 0 when the answer contains any string repeated more than 10 times. Otherwise, it rewards the model with a full score of 1.

As shown in Figure 17, the RLVR no-repetition reward improves the performance of Qwen2.5-Math-7B and Qwen2.5-Math-1.5B on MATH and AMC. We observe minimal or even negative improvement on other models. Based on our findings, we hypothesize that various patterns exist whose presence correlates with answer correctness. These patterns, including code reasoning and no repetition, can be easily elicited by RLVR even when the rewards provide no information about the ground-truth answers. Still, the effectiveness of eliciting these patterns is heavily model-dependent.

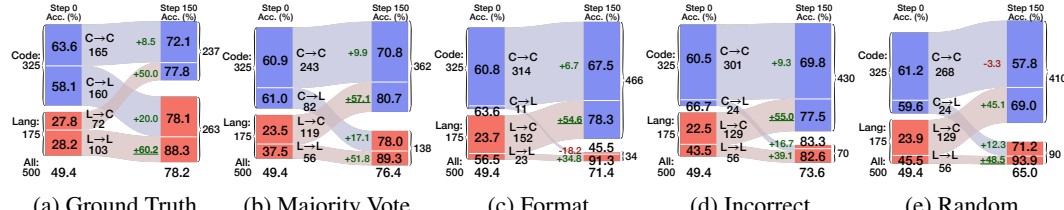

(a) Ground Truth  (b) Majority Vote  (c) Format  (d) Incorrect  (e) Random

Figure 18: Reasoning strategy switching and fine-grained performance of Qwen2.5-Math-7B on the MATH-500 test set before and after RLVR with different training signals. Blue labels are the problem for which the model uses code reasoning, while red labels indicate reasoning traces using only natural language. Accuracy of each disjoint subset of problems before and after RLVR is shown in the shaded ends, and the size of each subset is shown in the lightly shaded region along with the change in accuracy. For all weak and spurious rewards, the model tends to use more code reasoning after RLVR. There is a small proportion of originally-code-reasoning problems being switched to language reasoning (Code→Lang); a majority of originally-language-reasoning problems convert to code reasoning (Lang→Code), on which we see the most significant performance increase after RLVR.

# H  SWITCH OF REASONING STRATEGIES DURING RLVR

In this section, we present further analysis on the change of reasoning strategies during RLVR for Qwen2.5-Math and Qwen2.5 models.

**Reasoning strategy switches during RLVR.**  Qwen2.5-Math-7B's accuracy increases by an average of 23.5 absolute points across different training signals. To further break down this gain, we track the performance of models trained on each training signal across four disjoint subsets of the test prompts: (1) **Code→Code**: the model uses code reasoning both before and after RLVR; (2) **Code→Lang**: the model initially uses code reasoning but switches to natural language reasoning; (3) **Lang→Code**: the model initially uses natural language reasoning but switches to code reasoning; and (4) **Lang→Lang**: the model uses natural language reasoning both before and after RLVR. Specifically, we focus on two interconnected metrics: frequency and accuracy of each subset. To systematically quantify the contribution of each subset to the performance gain, we define a *Partial Contribution Score*, $C_d$, for any subset $d \subseteq \mathcal{D}$ of the entire test set $\mathcal{D}$, such that $C_d$ is the ratio between the net increase in the number of correctly answered problems in $d$ divided by the net increase in the number of correctly answered problems in $\mathcal{D}$:

$$C_d = \frac{\sum_{x \in d} \mathbb{I}[\text{correct}(x_t)] - \mathbb{I}[\text{correct}(x_0)]}{\sum_{x \in \mathcal{D}} \mathbb{I}[\text{correct}(x_t)] - \mathbb{I}[\text{correct}(x_0)]}, \quad \text{where } x_0 \text{ and } x_t \text{ are the initial and final answers.}$$

**Frequency:**  Figure 18 shows Qwen2.5-Math-7B's reasoning strategy switching pattern. For all weak and spurious rewards, the model uses more code reasoning after RLVR. While few originally-code-reasoning problems switch to language reasoning (Code→Lang), most originally-language-reasoning problems convert to code reasoning (Lang→Code). Ground truth reward does not follow this pattern. For *Bad-Code* models (Qwen2.5-7B and OLMo2-7B-SFT), meaningful rewards steer models away from bad code reasoning. Code reasoning decreases with ground truth, majority vote, and incorrect rewards for Qwen2.5-7B, but only with ground truth and majority vote for OLMo2-7B-SFT (Figures 5b and 5c). For *No-Code* models, RLVR fails to elicit meaningful changes in reasoning strategy, as this capability is likely not learned during pre-training.

**Accuracy:** From Figure 18, there is a drastic increase in accuracy in the Lang→Code subset after RLVR across all training signals. This is reflected in Table 3, which shows that 58.3% of the performance gain of Qwen2.5-Math-7B is from this subset. Similarly in Qwen2.5-Math-1.5B, switching from natural language reasoning to code reasoning contribute to 78.7% of the performance gain. For the *Bad-Code* models, Code→Lang contributes to 93.9% of the performance gain of Qwen2.5-7B. This is intuitive, as the model has a higher langauge reasoning accuracy than code reasoning accuracy, RLVR training essentially encourages the model to use the reasoning strategy that it is better at. For *No-Code* models, since there is no code reasoning before or after RLVR, all

Table 3: Partial contribution to the overall performance gain averaged over rewards that successfully steered the model's reasoning strategy (Figure 5).

| Model | Qwen2.5-Math-7B | Qwen2.5-Math-1.5B | Qwen2.5-7B |
|---|---|---|---|
| **Avg. Total Gain** | ↑ 23.5% | ↑ 28.5% | ↑ 30.6% |
| $C_{Code \to Code}$ | 11.6% | 2.8% | 0.2% |
| $C_{Code \to Lang}$ | 8.6% | 2.0% | **93.9%** |
| $C_{Lang \to Code}$ | **58.3%** | **78.7%** | 0.0% |
| $C_{Lang \to Lang}$ | 21.4% | 16.5% | 5.9% |

Table 4: Details of different prompts used in previous RLVR research. We explore the impact of existing prompts (Qwen Default, Simplerl-zoo, and Sober) and 2 our two proposed prompts—the first prompt, Math Prompt, indicates the evaluation domain without extra information about the format; the second prompt, Spurious Prompt, is a randomly picked LaTeX placeholder text generated by LIPSUM. Our motivation for introducing these 2 prompts is to study how concise domain knowledge or even random strings in context can boost the evaluation performance.

| Prompt Name | System Prompt | User Prompt |
|---|---|---|
| **Qwen Default** (Yang et al., 2024a) | Please reason step by step, and put your final answer within \boxed{}. | {} |
| **Math Problem** | You are a helpful Assistant. | Math Problem: {} |
| **Simplerl-zoo** (Zeng et al., 2025) | You are a helpful Assistant. | {}\nPlease reason step by step, and put your final answer within \boxed{}. |
| **Sober** (Hochlehnert et al., 2025) | Please reason step by step, and put your final answer within \boxed{}. [a] | Solve the following math problem efficiently and clearly. The last line of your response should be of the following format: 'Therefore, the final answer is: The last line of your response should be of the following format: 'Therefore, the final answer is: \boxed{ANSWER}. I hope it is correct' (without quotes) where ANSWER is just the final number or expression that solves the problem. Think step by step before answering.\n\n{} |
| **Spurious Prompt** | Lorem ipsum dolor sit amet, consectetuer adipiscing elit. | Ut purus elit, vestibulum ut, placerat ac, adipiscing vitae, felis. Curabitur dictum gravida mauris. Nam arcu libero, nonummy eget, consectetuer id, vulputate a, magna. Donec vehicula augue eu neque. Pellentesque habitant morbi tristique senectus et netus et malesuada fames ac turpis egestas. Mauris ut leo. Cras viverra metus rhoncus sem. Nulla et lectus vestibulum urna fringilla ultrices. Phasellus eu tellus sit amet tortor gravida placerat.\n\n{} |

[a]Note that the original paper did not include a system prompt. However, when applying chat templates to a conversation that lacks a system prompt (as occurs in the code from Hochlehnert et al. (2025)), the Qwen2.5-Math models automatically prepend their default system prompt.

performance gains (or losses) are from the **Lang→Lang** subset. These results suggest that much of the accuracy gain from RLVR on these spurious rewards in Qwen2.5-Math and Qwen2.5 is simply from eliciting the right reasoning strategy from the model.

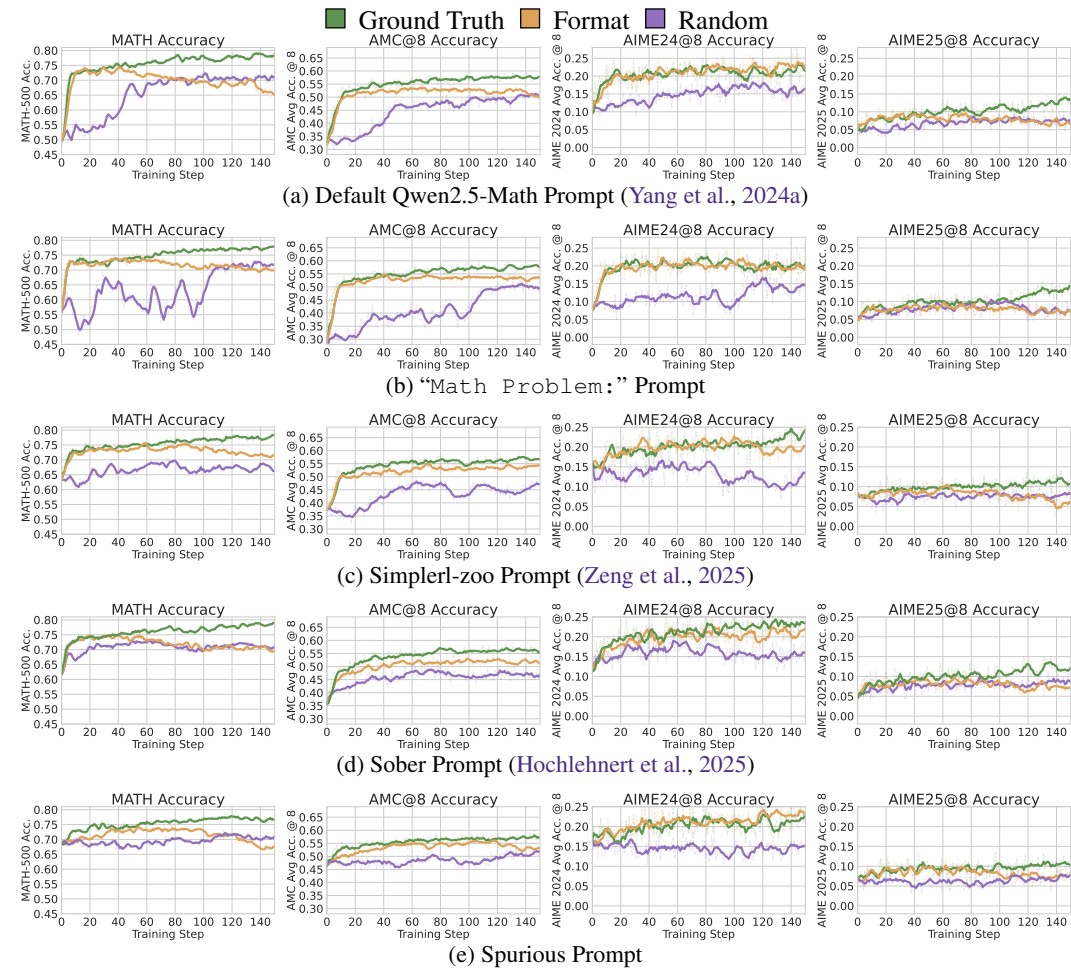

Figure 19: RLVR performance with different prompt templates on Qwen2.5-Math-7B. We show the choice of the prompt impacts both the before training performance and the RLVR training trajectories for Qwen2.5-Math-7B. We find the default prompt provided by Qwen2.5-Math (Yang et al., 2024a) results in lower initial performance; the other prompts offer higher initial performance, while Spurious Prompt offers the highest initial performance.

Table 5: Accuracy on MATH-500 and percent of parsable (format-following) responses on Qwen2.5-Math-7B with various prompts from Table 4. Even with a spurious prompt, the model is able to follow the format 84.1% of the time. It is not obvious that much of the performance can be explained by format-following.

|  | Default | MathProblem | SimpleRL-Zoo | Sober | Spurious |
|---|---|---|---|---|---|
| **MATH Acc.** | 49.4 | 55.8 | 63.2 | 61.60 | **68.8** |
| **% Parsable** | 78.9 | 72.1 | 85.4 | **93.1** | 84.1 |

## I    SPURIOUS PROMPTS: QWEN2.5-MATH-7B'S UNREASONABLY HIGH SENSITIVITY TO PROMPTS

As we show in Section 4.3, prompting the model to use code reasoning brings a 15.0% gain on Qwen2.5-Math-7B. This suggests that prompt engineering is a valid approach to elicit desired behaviors, which does not require parameter updates. In this section, we conduct an additional analysis on how choosing different existing prompts can impact model behavior and how such a

technique interacts with RLVR. In addition, we show that model performance can be improved even when using some task-unrelated, information-less prompts, which we name "spurious prompts."

## I.1 Existing Prompts

We collect different prompts from popular RL training frameworks and recent evaluation works: Qwen Default (Yang et al., 2024a), SimpleRL-Zoo (Zeng et al., 2025), Sober Reasoning (Hochlehnert et al., 2025), and our hand-constructed prompts (detailed in Table 4). Specifically, we introduce two new prompts—the first prompt, Math Prompt, indicates the evaluation domain without extra information about the format; the second prompt, Spurious Prompt, is a randomly picked LaTeX placeholder text generated by LIPSUM. Although it has been known that LLM evaluation can be sensitive to evaluation prompts (Sclar et al., 2024), our motivation to introduce these two particular prompts is to study how concise domain knowledge or even random perturbation in context can impact evaluation performance. We perform RL training using these prompts for both trajectory rollouts and evaluation.

As shown in Table 5, Sober prompt brings the highest parsable rate of the answers, while our spurious prompt leads to the highest accuracy on MATH-500. The results indicate that the model is very sensitive to prompts and the best performance does not necessarily require the highest parsable rate nor task-relevant information in context.

We further show the training trajectories using different prompts in Figure 19. We find that models trained with the Qwen default, MATH PROBLEM:, and Simplerl-zoo converge to similar performance after RLVR with the same training setting. We conjecture that the behavior that can be elicited by prompting tends to be a subset of the easy-to-elicit behaviors in RLVR, while RLVR can elicit additional behaviors that are not predefined in prompts.

## I.2 Spurious Prompts

As Qwen2.5-Math-7B shows high sensitivity to the prompts, we are curious about how much it could be impacted by spuriously unrelated prompts, e.g., the placeholder text in LaTex generated by LIPSUM. Specifically, we construct a spurious prompt using LIPSUM with a similar prompt length as the Sober prompt in Table 4. Surprisingly, the spurious prompt gives the highest initial performance compared with the other commonly used task-specific prompts, as shown in Figure 19e. Although we find that the improvement does not always happen for randomly picked prompts, the high performance with our spurious prompt indicates that the model can be very sensitive to in-context perturbations, where the benefit may sometimes originate from this sensitivity rather than from the task-relevant content in the prompt.

# J Additional Results on Models That Have Undergone RL Training

In this section, we provide additional results for models that have been trained with reinforcement learning, including two Qwen Instruct models (Qwen2.5-Math-7B-Instruct (Yang et al., 2024a) and Qwen2.5-7B-Instruct (Qwen et al., 2025)) and one Tulu3 model (Llama-3.1-Tulu-3-8B (Lambert et al., 2024)). In this section, we show that the post-RL models may behave differently from other base or instruction-tuned models in RLVR.

As shown in Figure 20, Qwen Instruct models show minimal improvement from RLVR training across different reward types in our setup, even when using ground-truth rewards. We observe this same pattern in other models: their performance plateaus after sufficient RLVR training steps. Therefore, we conjecture that Qwen2.5-Math-7B-Instruct benefits less from our RLVR experiments because it has already reached saturation from its previous RL training. While improved RLVR algorithms or higher-quality training data could still yield gains on the currently saturated models, we leave this exploration for future work.

Furthermore, we examine the RLVR behavior of Llama-3.1-Tulu-3-8B (Figure 20c) and find that, unlike the Qwen2.5 Instruct models, training with ground truth rewards provides only marginal gains on MATH and limited gains on ACM and AIME. The trends we observe for Llama-3.1-Tulu-3-8B

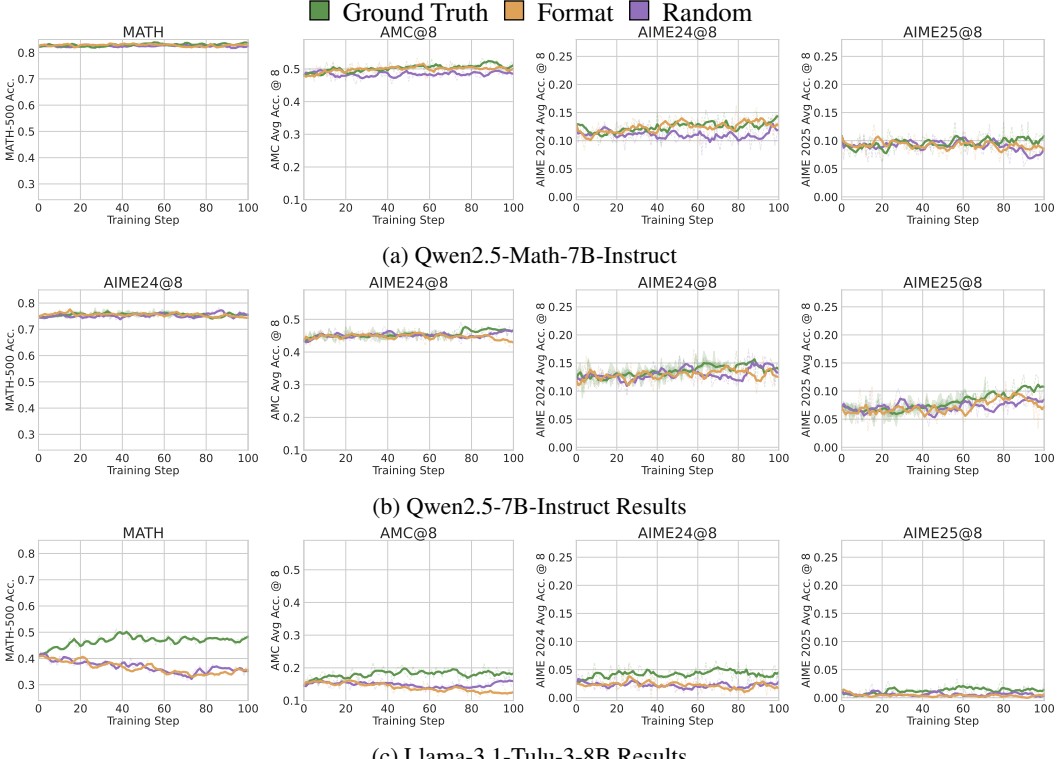

(a) Qwen2.5-Math-7B-Instruct

(b) Qwen2.5-7B-Instruct Results

(c) Llama-3.1-Tulu-3-8B Results

Figure 20: RLVR performance on models that have undergone RL training. We find that Qwen Instruct models show minimal improvement from our RLVR, while Llama-3.1-Tulu-3-8B exhibits similar patterns to OLMo models (Figure 3). There is a clear performance distinction when training with ground truth labels for Llama-3.1-Tulu-3-8B, but not for the Qwen Instruct models.

are similar to those we observe for its base model Llama-3.1-8B. We conjecture that this might be because these models share similar prior knowledge before RL training, which leads to similar RLVR behaviors. However, unlike the Qwen Instruct models, Llama-3.1-Tulu-3-8B still shows gains on MATH despite having been trained with RLVR on the Tulu3 dataset.

# K   ADDITIONAL RELATED WORKS

**Reinforcement Learning for Language Models.** The development of language model capabilities has been significantly advanced through reinforcement learning approaches. RLHF has become a standard technique for aligning models with human preferences (Ouyang et al., 2022; Bai et al., 2022), while RLVR has proven effective for tasks with deterministic answers (DeepSeek-Math, 2024; Gao et al., 2024; Wen et al., 2025; Song et al., 2025; Team et al., 2025; Lambert et al., 2024; Zeng et al., 2025; Luo et al., 2025a;b; Liu et al., 2025; Fatemi et al., 2025; He et al., 2025; Team, 2025; Wang et al., 2025b; Zhao et al., 2025a). These methods traditionally rely on accurate supervision signals, either through human feedback or verifiable rewards. Recent work has explored reducing the dependence on human annotations through AI feedback mechanisms (Bai et al., 2022) and through training dynamics analysis (Zhao et al., 2025b), which supports the finding that RL primarily amplifies behaviors or capabilities already buried in the pretrained models (Liu et al., 2025; Yue et al., 2025; Gandhi et al., 2025).

# L  QUALITATIVE ANALYSIS ON QWEN2.5-MATH-7B'S CODING REASONING BEHAVIORS

In this section, we show several qualitative examples on how Qwen2.5-Math-7B can reason in code. In addition, we show its code reasoning behavior is robust to numerical perturbations—the model generates similar code to solve the numerical perturbed question. However, we find Qwen2.5-Math-7B only uses natural language solutions when we rephrased the question using an alternative narrative.

**Qwen2.5-Math-7B can reason in code.**  In Figure 22–23, we show 3 qualitative examples of Qwen2.5-Math-7B outputs on 3 randomly picked questions from MATH-500. We find Qwen2.5-Math-7B is able to conduct coding reasoning, i.e., solving a problem by writing code, and predict the code execution outputs. Surprisingly, the model can predict the code execution outputs with a relatively high accuracy—in the examples shown in Figure 22 and Figure 21, Qwen2.5-Math-7B is able to compute the execution answers with 16-float precision **without** access to any code interpreter. In the example shown in Figure 23, Qwen2.5-Math-7B gives a wrong answer 4323. However, it is still very close to the ground-truth answer 4343.

**Qwen2.5-Math-7B's code reasoning behavior is robust to numerical perturbations.**  We further perturbed the number 999 in the question shown in Figure 23 to a few random integer numbers between the range of 100 and 1000, as shown in Figure 24–26. We find that Qwen2.5-Math-7B is capable of solving our perturbed questions with similar code snippets in their answers. For example, in Figure 24, we show that Qwen2.5-Math-7B is able to derive the correct answer 4344 in the numerically perturbed version of the question in Figure 23—we changed the input number from 999 to 1000, and the corresponding ground-truth answer changed from $4343_6$ to $4344_6$. We find Qwen2.5-Math-7B used the same code function to solve the question, and interestingly, it predicts the correct answer for the perturbed question, although it predicted an inaccurate answer for the original question in Figure 23.

**Qwen2.5-Math-7B's code reasoning behavior may not directly generalize to rephrased questions.**
We also rephrased the question in Figure 23, Figure 25, and Figure 26, respectively, with an alternative narrative. As shown in Figure 27–29, Qwen2.5-Math-7B first identifies and concludes the same core question in the first sentence of its answer. However, it does not use any code to answer this rephrased question. We hypothesize that whether Qwen2.5-Math-7B uses code reasoning to answer the question depends on the specific prompt distribution, i.e., whether the question is close to any of its training data that is augmented with code solutions, rather than the problem itself.

**Original MATH Question:**

Convert the point $(0, 3)$ in rectangular coordinates to polar coordinates. Enter your answer in the form $(r, \theta)$, where $r > 0$ and $0 \leq \theta < 2\pi$.

**Qwen2.5-Math-7B Solution (correct):**

To convert the point $(0, 3)$ in rectangular coordinates to polar coordinates $(r, \theta)$, we need to use the following formulas:
1. The distance $r$ from the origin to the point:

$$r = \sqrt{x^2 + y^2}$$

where $x = 0$ and $y = 3$.
2. The angle $\theta$ formed with the positive x-axis:
For the point $(0, 3)$:

- $r = \sqrt{0^2 + 3^2} = 3$

- Since $x = 0$ and $y = 3$, we are in the case where $x = 0$ and $y > 0$, so $\theta = \frac{\pi}{2}$.

Let's write the Python code to confirm these calculations and output the polar coordinates in the desired format.

```python
import math
# Rectangular coordinates
x = 0
y = 3
# Calculate r
r = math.sqrt(x**2 + y**2)
# Calculate θ
if x == 0 and y > 0:
    theta = math.pi / 2
elif x == 0 and y < 0:
    theta = 3 * math.pi / 2
else:
    theta = math.atan2(y, x)
# Ensure θ is in the range [0, 2π)
if theta < 0:
    theta += 2 * math.pi
print(f"r={r},θ = {theta}")
```

output: r = 3.0, $\theta$ = 1.5707963267948966
The polar coordinates for the point $(0, 3)$ are $(3, \frac{\pi}{2})$.
Thus, the final answer in the desired format is:

$(3, \frac{\pi}{2})$

Figure 21: Qualitative example of Qwen2.5-Math-7B outputs on one randomly picked question from MATH-500. Note that both the code and the corresponding code execution result are generated by Qwen2.5-Math-7B. **We do NOT provide any external code interpreter to the model.**

**Original MATH Question:**

What is the distance, in units, between the points $(2, -6)$ and $(-4, 3)$? Express your answer in simplest radical form.

**Qwen2.5-Math-7B Solution (correct):**

To find the distance between two points $(x_1, y_1)$ and $(x_2, y_2)$ in a Cartesian plane, we use the distance formula:

$$d = \sqrt{(x_2 - x_1)^2 + (y_2 - y_1)^2}$$

Given the points $(2, -6)$ and $(-4, 3)$, we can substitute these values into the formula:

$$d = \sqrt{((-4) - 2)^2 + (3 - (-6))^2}$$

Let's break this down step-by-step and compute the result using Python.

```python
import math

# Coordinates of the points
x1, y1 = 2, -6
x2, y2 = -4, 3

# Calculate the differences
dx = x2 - x1
dy = y2 - y1

# Calculate the distance using the distance formula
distance = math.sqrt(dx**2 + dy**2)
print(distance)
```

output: 10.816653826391969

The distance between the points $(2, -6)$ and $(-4, 3)$ is approximately $10.82$ units. To express the answer in simplest radical form, we need to check if this value can be simplified.

The distance formula calculation is:

$$d = \sqrt{((-4) - 2)^2 + (3 - (-6))^2} = \sqrt{(-6)^2 + (9)^2}$$
$$= \sqrt{36 + 81} = \sqrt{117} = \sqrt{9 \cdot 13} = 3\sqrt{13}$$

So, the exact distance in simplest radical form is $3\sqrt{13}$.

Thus, the final answer is:

$$\boxed{3\sqrt{13}}$$

Figure 22: Qualitative example of Qwen2.5-Math-7B outputs on one randomly picked question from MATH-500. Note that both the code and the corresponding code execution result are generated by Qwen2.5-Math-7B. **We do NOT provide any external code interpreter to the model.**

**Original MATH Question:**

What is the base six equivalent of $999_{10}$?

**Owen2.5-Math-7B Solution (incorrect, ground truth: $4343_6$):**

To convert a decimal number to its base-six equivalent, we repeatedly divide the number by 6 and keep track of the remainders. The remainders, read in reverse order, give the base-six representation of the number.

Here's the step-by-step process to convert $999_{10}$ to base six:

1. Divide the number by 6 and record the remainder.

2. Update the number to the quotient of the division.

3. Repeat steps 1 and 2 until the quotient is 0.

4. The base-six representation is the sequence of remainders read from bottom to top.

Let's implement this in Python to get the accurate result:

```python
def decimal_to_base_six(n):
    if n == 0:
        return "0"

    base_six_digits = []
    while n > 0:
        remainder = n % 6
        base_six_digits.append(str(remainder))
        n = n // 6

    # The base-six representation is the reverse of the
        collected remainders
    base_six_digits.reverse()
    return "".join(base_six_digits)

# Convert 999 from decimal to base six
decimal_number = 999
base_six_representation = decimal_to_base_six(decimal_number)
print(base_six_representation)
```

4323

The base-six equivalent of $999_{10}$ is $\boxed{4323}$.

Figure 23: Qualitative example of Qwen2.5-Math-7B outputs on one randomly picked question from MATH-500. Note that both the code and the corresponding code execution result are generated by Qwen2.5-Math-7B. **We do NOT provide any external code interpreter to the model.**

**Numerically Perturbed MATH Question:**

What is the base six equivalent of $1000_{10}$?

**Qwen2.5-Math-7B Solution (correct):**

To convert a decimal number (base 10) to a base 6 number, we repeatedly divide the number by 6 and keep track of the remainders. The base 6 number is then composed of the remainders read in reverse order. Let's perform this conversion step by step.
1. Divide the number by 6, record the remainder, and replace the number with the quotient. 2. Repeat until the quotient is 0. 3. The base 6 number is the sequence of remainders read in reverse order.
Let's do this process using Python code.

```python
def decimal_to_base_six(n):
    if n == 0:
        return '0'

    base_six_digits = []
    while n > 0:
        remainder = n % 6
        base_six_digits.append(str(remainder))
        n = n // 6

    # The remainders give the base 6 number in reverse order
    base_six_digits.reverse()
    return ''.join(base_six_digits)

# Convert 1000 from base 10 to base 6
base_six_number = decimal_to_base_six(1000)
print(base_six_number)
```

4344
The base six equivalent of $1000_{10}$ is $\boxed{4344_6}$.

Figure 24: Qualitative example of Qwen2.5-Math-7B outputs on the numerically perturbed MATH-500 question as shown in Figure 23. Note that both the code and the corresponding code execution result are generated by Qwen2.5-Math-7B. **We do NOT provide any external code interpreter to the model.**

**Numerically Perturbed MATH Question:**

What is the base six equivalent of $998_{10}$?

**Qwen2.5-Math-7B Solution (correct):**

To convert the decimal number $998_{10}$ to its base six equivalent, we need to repeatedly divide the number by 6 and keep track of the remainders. The base six digits are the remainders read from bottom to top (from last division to first).
Here is the step-by-step process:

1. Divide the number by 6.

2. Record the remainder.

3. Update the number to the quotient of the division.

4. Repeat steps 1-3 until the quotient is 0.

5. The base six number is the sequence of remainders read from bottom to top.

Let's do this step-by-step for $998_{10}$:

$$998 \div 6 = 166 \text{ remainder } 2 \tag{2}$$
$$166 \div 6 = 27 \text{ remainder } 4 \tag{3}$$
$$27 \div 6 = 4 \text{ remainder } 3 \tag{4}$$
$$4 \div 6 = 0 \text{ remainder } 4 \tag{5}$$

The remainders, read from bottom to top, give us the base six number. So, $998_{10}$ in base six is $4342_6$.
We can verify this by converting back to decimal:

$$4 \cdot 6^3 + 3 \cdot 6^2 + 4 \cdot 6^1 + 2 \cdot 6^0 = 4 \cdot 216 + 3 \cdot 36 + 4 \cdot 6 + 2 \tag{6}$$
$$= 864 + 108 + 24 + 2 \tag{7}$$
$$= 998_{10} \tag{8}$$

Now, let's implement this in Python to ensure the accuracy.

```python
def decimal_to_base_six(n):
    if n == 0:
        return "0"
    base_six_digits = []
    while n > 0:
        remainder = n % 6
        base_six_digits.append(str(remainder))
        n = n // 6
    base_six_digits.reverse()
    return ''.join(base_six_digits)

# Convert 998 from base 10 to base 6
base_six_number = decimal_to_base_six(998)
print(base_six_number)

4342
```

The base six equivalent of $998_{10}$ is $\boxed{4342_6}$.

Figure 25: Qualitative example of Qwen2.5-Math-7B outputs on the numerically perturbed MATH-500 question as shown in Figure 23. Note that both the code and the corresponding code execution result are generated by Qwen2.5-Math-7B. **We do NOT provide any external code interpreter to the model.**

**Numerically Perturbed MATH Question:**

What is the base six equivalent of $100_{10}$?

**Qwen2.5-Math-7B Solution (correct):**

To convert a decimal number (base 10) to a base-six number, we can repeatedly divide the number by 6 and keep track of the remainders. The conversion process involves the following steps:

1. Divide the number by 6.
2. Record the remainder.
3. Update the number to the quotient from the division.
4. Repeat the process until the quotient is 0.
5. The base-six number is the sequence of remainders read in reverse order.

Let's convert $100_{10}$ to base six using Python.

```python
def decimal_to_base_six(n):
    if n == 0:
        return "0"

    base_six_digits = []

    while n > 0:
        remainder = n % 6
        base_six_digits.append(str(remainder))
        n = n // 6

    # The base-six number is the sequence of remainders read
    # in reverse order
    base_six_digits.reverse()
    return ''.join(base_six_digits)

# Convert 100 (decimal) to base six
decimal_number = 100
base_six_number = decimal_to_base_six(decimal_number)
print(base_six_number)
```
```
244
```

The base six equivalent of $100_{10}$ is $\boxed{244_6}$.

Figure 26: Qualitative example of Qwen2.5-Math-7B outputs on the numerically perturbed MATH-500 question as shown in Figure 23. Note that both the code and the corresponding code execution result are generated by Qwen2.5-Math-7B. **We do NOT provide any external code interpreter to the model.**

**Rephrased MATH Question in an Alternative Narrative:**

If I have 999 objects in standard decimal counting, and I need to express this quantity using only the digits 0-5 in a positional numeral system, what sequence of digits would represent this amount?

**Qwen2.5-Math-7B Solution 2 (correct):**

To represent 999 using only the digits 0-5, we need to convert it to a base 6 number system. We need to divide this number by the powers of 6 and take the remainder until we run out of powers of 6.

999 divided by $6^3$ (which is 216) is 4 with a remainder of 135.

135 divided by $6^2$ (which is 36) is 3 with a remainder of 27.

27 divided by $6^1$ (which is 6) is 4 with a remainder of 3.

3 divided by $6^0$ (which is 1) is 3 with no remainder.

So, 999 in base 6 is $4343_6$.

$$4 \times 6^3 + 3 \times 6^2 + 4 \times 6^1 + 3 \times 6^0 = 4 \times 216 + 3 \times 36 + 4 \times 6 + 3 \times 1$$
$$= 864 + 108 + 24 + 3$$
$$= 999$$

Therefore, the representation of 999 using only digits 0-5 in a base-6 numeral system is $4343_6$.

Figure 27: Qualitative example of Qwen2.5-Math-7B outputs on the MATH-500 question shown in Figure 23 but with an alternative narrative. Qwen2.5-Math-7B uses natural language to answer the rephrased question without code reasoning.

---

### Rephrased MATH Question in an Alternative Narrative:

If I have 998 objects in standard decimal counting, and I need to express this quantity using only the digits 0-5 in a positional numeral system, what sequence of digits would represent this amount?

---

### Qwen2.5-Math-7B Solution 2 (correct):

To express the quantity of 998 using only the digits 0-5 in a positional numeral system, we need to convert 998 to a base-6 number. Here's the step-by-step process:

1. **Determine the largest power of 6 less than or equal to 998:** - $6^0 = 1$ - $6^1 = 6$ - $6^2 = 36$ - $6^3 = 216$ - $6^4 = 1296$ (which is greater than 998)
So, the largest power of 6 less than or equal to 998 is $6^3 = 216$.

2. **Find the coefficient for $6^3$:** - $998 \div 216 \approx 4.62$ - The integer part is 4, so the coefficient for $6^3$ is 4. - Subtract $4 \times 216 = 864$ from 998 to get the remainder: $998 - 864 = 134$.

3. **Find the coefficient for $6^2$:** - $134 \div 36 \approx 3.72$ - The integer part is 3, so the coefficient for $6^2$ is 3. - Subtract $3 \times 36 = 108$ from 134 to get the remainder: $134 - 108 = 26$.

4. **Find the coefficient for $6^1$:** - $26 \div 6 \approx 4.33$ - The integer part is 4, so the coefficient for $6^1$ is 4. - Subtract $4 \times 6 = 24$ from 26 to get the remainder: $26 - 24 = 2$.

5. **Find the coefficient for $6^0$:** - The remainder is 2, so the coefficient for $6^0$ is 2.

Putting it all together, the base-6 representation of 998 is $4342_6$.

Therefore, the sequence of digits that represents 998 in a base-6 positional numeral system is $\boxed{4342}$.

Figure 28: Qualitative examples of Qwen2.5-Math-7B outputs on the MATH-500 question shown in Figure 23 but with an alternative narrative. Qwen2.5-Math-7B uses natural language to answer the rephrased question without code reasoning.

**Rephrased MATH Question in an Alternative Narrative:**

If I have 100 objects in standard decimal counting, and I need to express this quantity using only the digits 0-5 in a positional numeral system, what sequence of digits would represent this amount?

**Qwen2.5-Math-7B Solution 1 (incorrect, ground truth: 244):**

To represent the number 100 using only the digits 0-5, you can use a base-6 numeral system. In a base-6 system, each digit represents a power of 6, starting from the right with $6^0$. The number 100 in base-10 can be converted to base-6 as follows:

$$
\begin{aligned}
100 &= 2 \times 6^2 + 0 \times 6^1 + 4 \times 6^0 \\
&= 2 \times 36 + 0 \times 6 + 4 \times 1 \\
&= 72 + 0 + 4 \\
&= 76
\end{aligned}
$$

However, since we are using base-6, the correct representation of 100 in this system is 204 ($2 \times 6^2 + 0 \times 6^1 + 4 \times 6^0$).

Figure 29: Qualitative examples of Qwen2.5-Math-7B outputs on the MATH-500 question shown in Figure 23 but with an alternative narrative and numerical perturbation. Qwen2.5-Math-7B uses natural language to answer the rephrased question without code reasoning. The equation expansion step in the solution is incorrect, leading to a final wrong answer: the calculation claims that $100 = 2 \times 6^2 + 0 \times 6^1 + 4 \times 6^0 = 72 + 0 + 4 = 76$, which is mathematically inconsistent.

