# OpenReview forum: "Spurious Rewards: Rethinking Training Signals in RLVR"
_ICLR.cc/2026/Conference — ICLR 2026 Conference Desk Rejected Submission_

### Official Review · Reviewer_duSx · 2025-10-28

**Soundness:** 2
**Presentation:** 2
**Contribution:** 2
**Rating:** 2
**Confidence:** 4

**Summary:**

This paper primarily finds that training with spurious rewards in reinforcement learning can also improve the performance of specific models on math reasoning tasks. The authors emphasize that this phenomenon mainly occurs in Qwen series models, while for other series of models such as Llama3.1-8b or Olmo2-7B, this phenomenon does not exist. The authors attempt to explain this discrepancy from whether different models possess the reasoning pattern of code reasoning, and find that code frequency can significantly improve the model's test performance.

**Strengths:**

The authors conducted RLVR experiments on multiple models and with different signals including Ground Truth, Majority Vote, One-Shot RL, Format Reward, Incorrect Label, and Random Reward, and the training process was comprehensively demonstrated.

**Weaknesses:**

I greatly admire the authors' attempt to explain why spurious rewards are only effective on Qwen series models through reasoning behaviors learned during pre-training (such as code reasoning). However, I am not convinced by their experimental design, observations, and corresponding explanations. The phenomena and hypotheses feel quite superficial, and some many observations and conclusions may be contradictory.


1. The authors propose in Section 4.2 that "Performance is correlated with code reasoning frequency," but from Figure 5, we can see that this conclusion only holds for Qwen2.5-math. For other models such as Qwen 2.5-7B and OLMo2-7B, we can clearly see that the conclusion is reversed. Although the authors explain that "for the Bad-Code models, we find that the reduction of code reasoning frequency is highly correlated to performance gains," I think this is an empirical observation lacking deep analysis, making it difficult to form a generalizable conclusion.

2. Moreover, the authors' conclusion that "RLVR with spurious rewards occur on the subset of questions where the model transitions from text-based to code-based reasoning after training" is arrived at too hastily, lacking rigorous theory or experimental design. It is important to emphasize that many experiments show correlation, which does not imply causation. We still hope to see rigorous theoretical or experimental proof.

3. The authors' statements in Lines 176-180 and Lines 358-362 may contradict each other. The former states that the model has not been taught new reasoning capabilities, but merely triggers latent ones already existing in the base model; the latter states that code reasoning declines and natural language reasoning declines, indicating that the model is learning natural language reasoning through RLVR's high-quality, ground truth rewards.

4. The authors' construction of incorrect rewards seems unfair. They first use majority voting, then filter out incorrect labels for training. However, the authors have not provided a reasonable explanation or conducted ablation studies or comparisons with randomly constructed incorrect rewards. Correspondingly, the conclusions the authors reach in Lines 415-419 seem less credible.

**Questions:**

1. Code reasoning is not the only pattern; other patterns such as reflection are also important reasoning behaviors. Can the authors also monitor these metrics during the training process?

2. Qwen models may have undergone pre-training on a large number of math reasoning tasks, leading to biased conclusions. Maybe authors could extend their conclusions to other tasks such as code, general science, or agentic tasks.

---

> ### Author Response · Authors · 2025-12-04
>
> We thank the reviewer for their thoughtful engagement with our work. We address each concern below and believe the clarifications resolve the perceived contradictions.
>
>
>
> ### **W1: Code reasoning correlation and model differences**
>
> We respectfully disagree with the characterization that our correlation claim "only holds for Qwen2.5-Math."  (Reviewer duSx) We **explicitly discuss all models** in Section 4.2 and note correlations in **both directions**:
>
> **Lines 363-366** state: *"For the Bad-Code models, we find that the **reduction of code reasoning frequency is highly correlated to performance gains**."*
>
> This is visible in Figure 5(b) for Qwen2.5-7B and Figure 5(c) for OLMo2-7B-SFT, where code frequency decreases as performance increases. **We present this as correlation for all models**—positive correlation for Good-Code models, negative correlation for Bad-Code models.
>
> The reviewer states "this conclusion is reversed" for other models—this is precisely what we report and is central to our argument. The **direction** of correlation depends on whether code reasoning is effective for that model (Table 1):
>
>
>
> * Good-Code (Qwen2.5-Math): Code → 60.9% acc, Lang → 35.0% acc → **positive** correlation
> * Bad-Code (Qwen2.5-7B): Code → 39.9% acc, Lang → 61.5% acc → **negative** correlation
> * No-Code (Llama, OLMo2-7B): No code generated → no correlation
>
> **This is not a contradiction but our core finding**: RLVR amplifies model-specific pre-existing behaviors. Whether increased code frequency helps or hurts depends on the model's pre-training.
>
> **Regarding causality**: Section 4.2 presents correlational evidence; Section 4.3 investigates causality through interventions (see W1 response above). We explicitly state (Line 369): *"Here, we investigate the **causal impact** of code reasoning..." *
>
> ## **W2: Correlation or Causation; lacks rigorous proof**
>
> We respectfully disagree that we lack rigorous experimental design for causation. We conduct multiple interventional experiments to control the frequency of code reasoning, and thereby study its causal impact on downstream performance in §4.3 and Appendix F that go beyond correlation to establish causal effects. Specifically, we provide:
>
>
>
> 1. **Prompting intervention** (Section 4.3, Table 2): Forcing code reasoning via prompt manipulation
> 2. **RL-based intervention** (Section 4.3, Figure 6): Python reward to induce code reasoning
> 3. **Suppression intervention** (Appendix F, Figure 16): Compound rewards to inhibit code reasoning
>
> Additionally, to explain *why* spurious rewards elicit code reasoning, we provide:
>
>
>
> 4. **Mechanistic ablation** (Figure 7): Demonstrating that clipping bias concentrates probability on high-frequency behaviors (including code reasoning)
> 5. **Theoretical analysis** (Appendix B.1.2): Formal gradient derivation showing clipping amplifies existing high-probability patterns
>
> These experiments demonstrate causal effects through direct manipulation and bidirectional validation (both inducing and suppressing the hypothesized cause). We elaborate on each below:

---

> > ### Author Response · Authors · 2025-12-04
> >
> > ### **Our Causal Evidence:**
> >
> > **Experiment 1: Prompting Intervention (Section 4.3, Table 2).** We directly manipulate code reasoning frequency by forcing models to begin responses with "Let's solve this using Python." If code reasoning causally improves performance, we should observe improvements specifically for models with strong code reasoning priors and degradation for models without such priors. The results confirm this prediction: Qwen2.5-Math-7B improves by +15.0%, Qwen2.5-Math-1.5B improves by +24.2%, while Llama3.2-3B-Instruct degrades by −28.6% and OLMo2-7B-SFT degrades by −2.8%. This bidirectional effect of helping models with good code priors while hurting those without provides strong evidence that code reasoning causally affects performance rather than merely correlating with it.
> >
> > **Experiment 2: Python Reward (Section 4.3, Figure 6).** In this experiment, we induce increased code reading frequency using RLVR. We train models using RLVR where the reward equals 1 if and only if the response contains the string "python." This intervention increases code reasoning frequency to > 99% after 20 training steps for Qwen2.5-Math-7B. If code reasoning causally improves performance, this intervention should improve Qwen models but not others. We observe that Qwen2.5-Math models show substantial performance gains, while other model families demonstrate minimal improvement, consistent with our causal hypothesis.
> >
> > **Experiment 3: Suppression via Compound Rewards (Appendix F, Figure 16).** We construct compound rewards that intersect our original rewards with a no-Python reward, thereby discouraging code reasoning. If code reasoning causally drives Qwen2.5-Math improvements, suppressing it should reduce gains. We find that format reward combined with no-Python eliminates performance gains for Qwen2.5-Math-7B compared to format reward alone. Conversely, for Bad-Code models like OLMo2-7B-SFT (where code reasoning correlates with incorrectness), the compound reward improves performance by +8.9% compared to format reward alone. This bidirectional suppression effect further supports the causal interpretation.
> >
> > **Experiment 4: Mechanistic Ablation (Appendix B, Figure 7).** To understand why spurious rewards elicit code reasoning, we ablate the clipping mechanism in GRPO through three methods: directly disabling clipping, increasing mini-batch size to match rollout size, and reducing rollout size. If clipping bias causally drives the effectiveness of random rewards, removing clipping should eliminate performance gains. We find that random rewards produce approximately 21% performance gains with clipping enabled, but produce no robust improvement when clipping is disabled across all three ablation methods. This ablation isolates the specific mechanism through which spurious rewards operate.
> >
> > **Theoretical Foundation (Appendix B.1.2).** We provide formal gradient analysis demonstrating that GRPO clipping creates systematic bias toward high-probability behaviors even under random rewards. The expected gradient under random rewards with clipping is non-zero and favors amplification of existing model priors. This theoretical analysis explains why spurious rewards can work and is validated by our empirical ablations.
> > ### **W3: "Lines 176-180 vs 358-362 contradict each other"**
> >
> > We respectfully clarify that these statements describe different training regimes rather than contradictory claims.
> >
> > **Lines 176-180** conclude that RLVR with spurious rewards "does not teach models new reasoning capabilities, but instead triggers latent ones already present in the base model." This applies specifically to training with spurious rewards (random, incorrect labels, format-only), where the reward signal provides no information about correctness. Here, RLVR cannot teach new capabilities—performance gains come from surfacing pre-existing patterns learned during pretraining.
> >
> > **Lines 358-362** observe that with ground truth rewards, "the model is learning real knowledge from RLVR on high-quality, ground truth rewards," as evidenced by improving natural language reasoning. This applies to training with ground truth rewards, where the reward signal directly indicates correctness. Here, RLVR can both surface pre-existing patterns *and* refine them through actual learning.
> >
> > The key distinction is: spurious rewards enable only **elicitation** (surfacing existing capabilities), while ground truth rewards enable **elicitation plus learning** (surfacing and refining capabilities). Figure 5a supports this: under ground truth rewards, code frequency rises then falls as the model learns better strategies. Under spurious rewards, code frequency just increases monotonically without refinement.
> >
> > We will revise the manuscript to make this distinction more explicit.

---

> > > ### Author Response · Authors · 2025-12-04
> > >
> > > ### **W4: "Incorrect reward construction is unfair/needs ablation"**
> > >
> > > We appreciate this methodological question. Our construction (majority voting -> filter for incorrect labels) was designed to ensure incorrect labels stay within the model's natural output distribution. This tests whether RLVR can improve performance even when training on plausible-but-wrong answers the model might naturally generate.
> > >
> > > We thank the reviewer for suggesting an ablation with randomly selected incorrect answers. The suggested ablation of "randomly constructed incorrect rewards" would assign arbitrary incorrect answers as targets. At each training step, model rollouts would almost never match these arbitrary targets, resulting in uniformly zero rewards and **zero variance in advantages**. This would eliminate the gradient signal in GRPO, which breaks model training.
> > >
> > >
> > > ---
> > >
> > >
> > > ## **Questions**
> > >
> > >
> > > ### **Q1: "Can authors monitor other patterns?"**
> > >
> > > We thank the reviewer for this suggestion. We respectfully clarify that we do not claim code reasoning is the only pattern contributing to spurious reward effectiveness. Appendix G already analyzes non-repetitive generation (Figure 17), which shows similar model-specific effects—it helps Qwen2.5-Math but not other models. We explicitly frame code reasoning as an "illuminating case study" (Line 270) precisely because it is operationally clear and easy to measure automatically.
> > >
> > > The paper examines multiple patterns (code reasoning and no-repetition), with code reasoning featured as the primary case study due to its measurement clarity. We acknowledge that additional patterns like step-by-step structure, worked examples, and subgoal decomposition could provide further insights. We also note the methodological challenges in detecting certain patterns—for example, "reflection" is conceptually important but difficult to operationalize automatically, unlike code which has clear string markers like "python".
> > >
> > > We welcome suggestions for operationalizing the detection of reflection or other behaviors.
> > >
> > >
> > > ### **Q2: "Extend to other domains (code, science, agentic tasks)"**
> > >
> > > We thank the reviewer for this excellent suggestion. We believe our work provides a comprehensive analysis within the mathematical reasoning domain, examining multiple models (10 model variants), multiple benchmarks (MATH-500, AMC, AIME 2024/2025), and multiple reward types (6 different reward signals).
> > >
> > > Our core contributions, demonstrating that spurious rewards can improve performance through elicitation of pre-existing capabilities, identifying code reasoning as a key mechanism for Qwen models, and providing causal intervention experiments, are well-supported within this domain. The mathematical reasoning setting is particularly well-suited for studying RLVR because it offers verifiable rewards and has been the primary focus of recent RLVR research.
> > >
> > > Cross-domain validation represents an interesting direction for future work. Extending our analysis to other domains such as code generation, scientific reasoning, or agentic tasks would be valuable for understanding whether similar elicitation mechanisms operate across different task types and whether other domain-specific patterns (beyond code reasoning) can be identified.
> > >
> > > We note that our methodological contribution—demonstrating that RLVR effectiveness is highly model-dependent and cautioning the community against drawing conclusions from single-model studies—is important for RLVR research practices in mathematical reasoning and potentially beyond.

---

### Official Review · Reviewer_K7Z4 · 2025-10-28

**Soundness:** 2
**Presentation:** 2
**Contribution:** 2
**Rating:** 2
**Confidence:** 4

**Summary:**

This paper shows that RLVR can significantly improve reasoning performance even with random or incorrect rewards, especially for Qwen2.5-Math models.  More specifically, the authors find that spurious rewards (random, format-based, or wrong answers) yield nearly the same gains as ground-truth rewards on MATH-500 and AMC. The effect, however, does not generalize to other model families like Llama3 or OLMo2. Based on the finding, they trace the improvement to Qwen’s pre-existing “code reasoning” behavior and also analysis the clipping bias in GRPO.

**Strengths:**

1. The paper conducts a very detailed set of experiments across different model families, including Qwen and Llama, which allows it to comprehensively demonstrate the scope of its conclusions, mainly applicable to the Qwen series.

2. In the final part, the paper provides an algorithmic analysis that explains its findings from the optimization perspective.

**Weaknesses:**

1. The experimental results in this paper lack sufficient justification. In Appendix I, the authors show Qwen2.5-Math-7B base model scores under different prompts: the best prompt (from SimpleRL-Zoo) reaches 63.2, while the prompt actually used in the main experiments scores only 49.4. This is a large and critical gap, yet Figure 1, the paper’s most important result, uses the lower 49.4 baseline. The same issue appears for Qwen2.5-7B, whose baseline is also lower than that reported in SimpleRL-Zoo. Such a discrepancy could substantially affect the conclusions. For example, some reward signals might actually decrease performance rather than increase it as shown in Figure 1. The choice of a lower baseline therefore requires a more detailed explanation and justification.

2. In Section 4, the connection to the main storyline is unclear. The paper claims that Qwen models tend to produce Python code while others either do not or produce incorrect code, and attributes this to pretraining data differences. However, this does not explain why spurious rewards benefit Qwen but not other models, nor how these rewards are linked to code reasoning. If, as the paper states, “RLVR with spurious rewards upweights pre-existing reasoning strategies,” then why can’t those strategies be similarly activated and improved in other models?

3. Similarly, in Section 5, the paper argues that the GRPO clipping bias explains why random rewards lead to improvement. Yet the same question remains: if clipping bias is the true cause, why don’t other base models also improve? In theory, all models should benefit from this optimization bias. Therefore, this conclusion may not capture the essential mechanism behind the observed phenomenon.

**Questions:**

1. Why is the “incorrect reward”(Lines 152–155) constructed based on majority voting rather than a simpler and more direct design, such as simply inverting the ground-truth labels (i.e., rewarding wrong answers)? What is the motivation for this choice?

2. In Figure 3, the incorrect reward achieves higher performance than the random reward, which seems counterintuitive. Intuitively, deliberately rewarding wrong answers should be even worse than random noise. Could the authors provide an explanation or hypothesis for why this occurs?

3. Do the findings in this paper, particularly the effectiveness of spurious rewards, extend to other reasoning domains such as code generation? Have the authors conducted or considered experiments outside mathematics to verify this?

---

> ### Author Response · Authors · 2025-12-04
>
> We thank the reviewer for their detailed feedback and for highlighting our comprehensive experimental coverage across model families and our algorithmic analysis from an optimization perspective. We appreciate the recognition of our extensive empirical work. However, we believe there are some misunderstandings about our work's purpose and experimental design that we would like to clarify below.
>
>
>
> ## **W1: Prompt Choice and Baseline Performance**
>
> > "Figure 1, the paper's most important result, uses the lower 49.4 baseline... Such a discrepancy could substantially affect the conclusions. For example, some reward signals might actually decrease performance rather than increase it as shown in Figure 1."
>
> We respectfully believe this concern stems from a misunderstanding of our experimental design.
>
> **We use the official Qwen prompt template**, and our 49.4% baseline **exactly matches the results in the Qwen technical report** (Yang et al., 2024a). This is not arbitrary—it's the standard evaluation protocol used by multiple recent RLVR works (Zuo et al., 2025; Wang et al., 2025b; Zeng et al., 2025).
>
> The key principle in our experimental design is **consistency**: the prompt used before, during, and after training must be identical to accurately measure RLVR gains. A higher initial performance with a different prompt does not invalidate our findings—it would simply mean training from a different starting point. **The crucial observation is that spurious rewards produce gains regardless of the starting baseline.**
>
> Moreover, **we extensively discuss prompt sensitivity in Appendix I** (pages 31-32, Table 4, Figure 19, as the reviewer has mentioned). We show that while different prompts yield different baseline performance (ranging from 49.4% to 68.8%), **spurious rewards still produce substantial gains across all prompt configurations**. This actually *strengthens* our findings by demonstrating robustness to prompt variation.
>
> **We emphasize that our paper is not proposing spurious rewards as a method**—we are demonstrating that the community may have overestimated what RLVR methods are learning, since even random rewards yield large gains on certain models. This is a diagnostic contribution rather than a methodological one.
>
> ## **W2: Connection Between Code Reasoning and Spurious Rewards**
>
> > "If... RLVR with spurious rewards upweights pre-existing reasoning strategies, then why can't those strategies be similarly activated and improved in other models?"
>
> **We believe this question highlights an important clarification we should make more prominent.** Code reasoning is not merely about *generating* code—it must correlate with correctness to be beneficial. As we show in Section 4.1 and Table 1, Qwen2.5-Math-7B achieves **60.9% accuracy with code vs. 28.0% without**. Other models either:
> - **Never generate code** (No-Code models: Llama, OLMo2-7B)
> - **Generate code that hurts performance** (Bad-Code models: OLMo2-7B-SFT has 21.0% accuracy with code vs. 40.0% without)
>
> The mechanism is as follows: spurious rewards amplify high-probability behaviors through clipping bias (Section 5, Appendix B.1.3). For Qwen2.5-Math, this happens to be beneficial code reasoning learned during pretraining. For other models, there is no such beneficial pre-existing pattern to amplify.
>
> **Additionally, we note that code reasoning is not the only beneficial pattern.** In Appendix G, we explore repetition avoidance as another behavior that RLVR can elicit. We find that a "no-repetition reward" (which simply discourages string repetition) also improves Qwen2.5-Math performance while providing minimal gains for other models. This demonstrates that multiple pre-existing patterns can be surfaced, but only when they correlate with correctness in the base model.
>
> We further validate the code reasoning hypothesis with ablation studies (Section 4.3, Figure 16): **suppressing code reasoning reduces Qwen gains but improves Bad-Code models**, exactly as our hypothesis predicts.

---

> > ### Author Response · Authors · 2025-12-04
> >
> > ## **W3: Clipping Bias and Model Specificity**
> >
> > > "if clipping bias is the true cause, why don't other base models also improve?"
> >
> > We appreciate this question as it allows us to clarify an important point. As we state in the paper, clipping bias elicits high-prior behaviors in the model. Different models have different high prior behaviors due to their training data, model architecture, training procedures, etc. Thus, the behaviors being upweighted with the clipping bias is differently beneficial for the math task performance for each model. Helpful code reasoning is a high-prior behavior in the Qwen-Math family of models specifically, whereas high-prior behaviors in other based models are not beneficial to the task, therefore the performance does not improve in these other models. **The clipping bias operates on all models** (Appendix B.1.2), but **its effect depends critically on what behaviors get amplified**. As we state in Appendix B.1.3:
> >
> > *"The clipping bias is universal, but its impact on performance depends entirely on whether the model's dominant pre-existing behaviors happen to be effective reasoning strategies."*
> >
> > As stated in the response to W2, for Qwen2.5-Math, clipping bias increases the probability of code reasoning (which correlates with correctness). For other models, there is no such beneficial pattern to amplify, so the bias operates but does not improve performance. **We empirically demonstrate this in Figure 9**: token probability and code frequency both increase with clipping enabled, but remain flat without clipping.
> >
> >
> > ## **Q1: Why Majority Voting for Incorrect Reward?**
> >
> > > "Why is the 'incorrect reward' constructed based on majority voting rather than... simply inverting the ground-truth labels?"
> >
> > We thank the reviewer for raising this question. The design of our incorrect reward was motivated by our initial hypothesis that "teaching confidence matters more than teaching correctness." We wanted to test whether rewarding the model's most confident (but wrong) predictions would still yield gains. Majority voting identifies these high-confidence incorrect answers.
> >
> > We acknowledge that an alternative design, in which we could reward *any* incorrect answer rather than a specific majority-voted one, would also be interesting to explore. This would test whether RLVR simply needs *any* consistent signal (even if anti-correlated with correctness) versus whether the signal needs to align with the model's existing output distribution.
> >
> > We focused on majority-voted incorrect labels as our primary spurious reward design, but we agree that exploring this alternative would provide additional insight into the mechanisms at play. **We did test alternative constructions** (selecting the 1st, 2nd, or least most common incorrect answer from 64 rollouts) and found similar results that these spurious rewards can improve performance on the Qwen-Math family of models, suggesting that as long as the incorrect label is a probable model output, the specific selection method is not critical.
> >
> >
> > ## **Q2: Why Incorrect > Random Performance?**
> >
> > > "In Figure 3, the incorrect reward achieves higher performance than the random reward... Could the authors provide an explanation?"
> >
> > We address this in Section 5 (page 8), but we appreciate the opportunity to clarify further. It is a design choice that incorrect labels are derived from majority voting over the model's own outputs. This means the model is being rewarded for what it was already likely to generate, effectively making it more confident in its existing reasoning patterns. Even though the specific rewarded answers are incorrect, these reasoning patterns happen to sometimes lead to correct answers.
> >
> > In contrast, random rewards provide no such consistent signal—they don't systematically reinforce any particular reasoning strategy, which explains why they converge more slowly and less reliably than incorrect rewards.

---

> > > ### Author Response · Authors · 2025-12-04
> > >
> > > ## **Q3: Generalization to Other Domains**
> > >
> > > Thank you for this suggestion. Our current study is indeed scoped to mathematical reasoning, and we agree that this is an important limitation for interpreting the broader impact of spurious rewards. We chose math for two practical reasons: (1) it is the primary testbed for RLVR in prior work, including all the papers whose methods we compare against [2,3,4], and (2) it is one of the few domains with high-quality, programmatic verifiers and unambiguous ground-truth labels, which lets us precisely control and ablate the reward signal. Due to compute and time constraints, we were unable to add non-math RL experiments including such experiments in this submission, but we agree with the reviewer that extending our analysis to, e.g., code and symbolic reasoning tasks with suitable verifiers is an important direction for future work, and we will emphasize this limitation and planned extension more clearly in the revised version.
> > >
> > > [2] Zuo, Yuxin, et al. "Ttrl: Test-time reinforcement learning." arXiv preprint arXiv:2504.16084 (2025).
> > > [3] Wang, Yiping, et al. "Reinforcement learning for reasoning in large language models with one training example." arXiv preprint arXiv:2504.20571 (2025).
> > > [4] Zhao, Xuandong, et al. "Learning to reason without external rewards." arXiv preprint arXiv:2505.19590 (2025).
> > >
> > >
> > >
> > > ## **Summary**
> > >
> > > We believe the concerns raised by the reviewer stem primarily from misunderstandings about our paper's goals and contributions rather than fundamental weaknesses in the work itself. Our central contribution is demonstrating that **RLVR gains on certain models may reflect pre-existing capabilities rather than genuine learning from reward signals**. We are not proposing spurious rewards as a method, but using them as a diagnostic tool to understand what RLVR is actually doing, which is a distinction that we believe is important for the community.
> > >
> > > If we have successfully addressed the reviewer's concerns, we would greatly appreciate it if the reviewer would consider raising their score to reflect the paper's contributions to understanding RLVR training dynamics.

---

### Official Review · Reviewer_w4Pw · 2025-11-01

**Soundness:** 4
**Presentation:** 4
**Contribution:** 4
**Rating:** 8
**Confidence:** 4

**Summary:**

This paper shows that RLVR can elicit strong mathematical reasoning improvements even with spurious rewards that have little, no, or negative correlation with correct answers. The authors demonstrate that Qwen2.5-Math-7B achieves 21-27% absolute gains on MATH-500 using completely spurious rewards while other model families like Llama3 and OLMo2 fail to benefit from such spurious signals. They identify "code reasoning" as a distinctive Qwen behavior that increases from 65% to over 90% after RLVR, even with spurious rewards. The authors hypothesize that RLVR surfaces useful reasoning representations learned during pretraining rather than relying solely on reward signals. However, the exact mechanism remains unclear. These findings suggest that RLVR research should be validated across diverse model families rather than relying on a single model, as it is surprisingly easy to obtain significant but potentially misleading performance gains with spurious rewards on certain architectures.

**Strengths:**

**\[S1\] Novel and impactful findings**
The paper presents a striking discovery that RLVR can produce substantial performance improvements even with spurious rewards (Qwen). Furthermore, the insight that RLVR methods should be validated across diverse model families rather than relying on a single architecture is a valuable contribution that will benefit the broader research community in avoiding potentially misleading conclusions.

**\[S2\] Rigorous experimental design and analysis**
The experiments and analyses are well-structured and effectively support the authors' claims. The progression from main results to deeper investigation is logical and convincing.

**Weaknesses:**

**\[W1\]** While the lack of access to pretraining data makes deeper investigation challenging and understandable, the paper's reliance on phenomenon discovery and analysis through one representative behavior (code reasoning) leaves some mechanistic questions unanswered. Although the authors reasonably acknowledge this as future work, a deeper mechanistic explanation would have strengthened the contribution.

**\[W2\]** In Section 5 (Curious case), the authors provide additional analysis that supports their hypothesis on random rewards. However, they lack such analysis for incorrect rewards, only providing a hypothesis without comparable supporting experiments. Given that incorrect rewards actively provide misleading signals (rather than just noise), additional investigation into how RLVR still produces gains in this setting would have been valuable.

**\[W3\]** The paper would benefit from analyzing pass@K performance on problems that were unsolvable by the base model (pass@K \= 0 before RLVR). Examining whether RLVR with spurious rewards enables solving such problems, particularly under large K, would more clearly distinguish genuine reasoning expansion from amplification of existing capabilities.

**\[W4\]** The experiments focus exclusively on mathematical reasoning tasks. It remains unclear whether these findings generalize to other domains or reasoning task types. Investigating whether spurious reward effectiveness is a math-specific phenomenon or appears across diverse reasoning tasks (e.g., logical reasoning, commonsense reasoning, code generation) would strengthen the generalizability and impact of the findings.

**Questions:**

**\[Q1\]** Could the authors clarify the motivation behind introducing format reward and provide additional analysis? Given that the \\boxed{} expression format is already included in Qwen2.5-Math's system prompt (L144), it is unclear why this same design choice was applied uniformly to other model families. Was this intentional to test cross-model behavior, or could different format conventions have been more appropriate for Llama3 and OLMo2?

**\[Q2\]** When performing multiple rollouts on the same problem, does the frequency of code reasoning vary across samples, and if so, does this variation correlate with accuracy?

**\[Q3\]** The paper notes that Qwen2.5-Math engages in code reasoning even without a code interpreter. Have the authors analyzed the quality or validity of the generated code (e.g., whether it is executable, syntactically correct, or error-free)? Is there a correlation between code quality and problem-solving performance, or does RLVR surface code-like reasoning patterns regardless of actual code validity?

---

> ### Author Response · Authors · 2025-12-04
>
> We thank the reviewer for their encouraging comments, acknowledging that our paper is novel and impactful with comprehensive experiments. We address the reviewers’ questions below.
>
> **W1.** The paper's reliance on phenomenon discovery and analysis through one representative behavior (code reasoning) leaves some mechanistic questions unanswered.
>
> **A1.** We would like to clarify remind the reviewer that we have provided a theoretical explanation of why random rewards work, based on the optimization bias in GRPO (Section 4 and Appendix B). At a high level, we hypothesize that random rewards are effective because the clipping term in the GRPO loss biases optimization and further upweights the policy’s existing high-probability behaviors. Concretely, in our setup, random rewards reinforce existing positive behaviors such as code reasoning, as shown in Section 4. We verify the mechanism behind random rewards both empirically and theoretically in Appendix B. We will make our writing clearer to emphasize that we do have a mechanistic explanation in the corresponding sections.
>
> Spurious rewards comprises multiple different types of rewards, among which, the most surprising type is random rewards. Therefore, we extend a detailed explanation on the mechanism of random rewards in Appendix B and discuss the other rewards in the main paper. Although this hypothesis was introduced to explain random rewards, the same token-level bias can also affect training when using format or incorrect-label rewards. In addition, these other rewards can provide additional helpful supervision. For example, the format reward concentrates training on answers that successfully derive a final solution without the LM giving up or exceeding the maximum context length; the majority-voted reward makes the model more likely to produce confident answers, bringing pass@1 performance closer to majority-vote accuracy; and we provide further explanation for incorrect rewards in A2 in response to the second weakness. Therefore, we believe that we have provided a reasonable and substantive explanation beyond the empirical analysis.
>
> **W2.** In Section 5 (Curious case), the authors provide additional analysis that supports their hypothesis on random rewards. However, they lack such analysis for incorrect rewards, only providing a hypothesis without comparable supporting experiments. Given that incorrect rewards actively provide misleading signals (rather than just noise), additional investigation into how RLVR still produces gains in this setting would have been valuable.Lack of analysis for incorrect reward.
>
> **A2.** We thank the reviewer for acknowledging our analysis that random rewards can support our hypothesis. We would like to note that the “incorrect rewards” in our paper used the “majority-voted incorrect answers” as labels for verification. Among these answers, we find that some are partially correct (e.g., when the question asks for two numbers, the model gets one of them correct), or they provide correct answers but the verifier fails to convert them to match the ground truth. In both cases, training the model toward these “incorrect labels” leads to improvements. In addition, some majority-voted but completely incorrect answers are derived from partially correct reasoning traces; enhancing such traces can help concentrate the model’s reasoning around this partially correct answer, making the training less effective but still potentially helpful for downstream tasks, as discussed in Section 5. We will update the discussion with extended discussions.
>
> We also tried creating randomly generated incorrect labels, but RL does not provide any training signal in that case, as the reasoning traces can never reach them. Inspired by the reviewer, we agree that it would be interesting for future work to further investigate how training performance correlates with the model’s confidence in the incorrect labels and with the degree of incorrectness of the reasoning process. Due to limited time and compute resources during the rebuttal period, we are unable to supplement new RL experiments. We defer this to future work and thank the reviewer again for their insightful comments.

---

> > ### Author Response · Authors · 2025-12-04
> >
> > **W3.** Request for pass@K analysis on problems unsolved by the base model to distinguish genuine reasoning expansion from amplification of existing capabilities.
> >
> > **A3.** We thank the reviewer for the suggestion! Prior work has shown that, even when RLVR is trained with ground-truth correctness rewards, pass@K curves change relatively little: RLVR mainly improves low-K efficiency, while base models quickly catch up or even surpass RL-trained models at larger K [1]. Therefore, we instead use average@K as the primary metric we track during training, since pass@K may not give much helpful signal. Nonetheless, we will supplement the pass@K metric in future experiments.
> >
> > [1] Yue, Yang, et al. "Does reinforcement learning really incentivize reasoning capacity in llms beyond the base model?." arXiv preprint arXiv:2504.13837 (2025).Does Reinforcement Learning Really Incentivize Reasoning Capacity in LLMs Beyond the Base Model?
> >
> >
> >
> > **W4.** The experiments are limited to the math domain.
> >
> > **A4.** Thank you for this suggestion. Our current study is indeed scoped to mathematical reasoning, and we agree that this is an important limitation for interpreting the broader impact of spurious rewards. We chose math for two practical reasons: (1) it is the primary testbed for RLVR in prior work, including all the papers whose methods we compare against [2,3,4], and (2) it is one of the few domains with high-quality, programmatic verifiers and unambiguous ground-truth labels, which lets us precisely control and ablate the reward signal. Due to compute and time constraints, we were unable to add non-math RL experimentsinclude such experiments in this submission, but we agree with the reviewer that extending our analysis to, e.g., code and symbolic reasoning tasks with suitable verifiers is an important direction for future work, and we will emphasize this limitation and planned extension more clearly in the revised version.
> >
> > [2] Zuo, Yuxin, et al. "Ttrl: Test-time reinforcement learning." arXiv preprint arXiv:2504.16084 (2025).
> > [3] Wang, Yiping, et al. "Reinforcement learning for reasoning in large language models with one training example." arXiv preprint arXiv:2504.20571 (2025).
> > [4] Zhao, Xuandong, et al. "Learning to reason without external rewards." arXiv preprint arXiv:2505.19590 (2025).
> >
> >
> > **Q1** Could the authors clarify the motivation behind introducing format reward and provide additional analysis?
> >
> > **AQ1** We added the format reward purely out of curiosity, as it is a commonly used auxiliary reward. Format reward is commonly used but it's also spurious in the sense that it doesn't reward actually arriving at the correct answer; we wanted to disentangle how much improvement came from better format or instruction following vs improved reasoning. We explored applying format rewards for RL training before testing random rewards, which It is related to the model’s instruction-following ability, and we expected toexpected it to work better than random rewards but weaker than majority-voted rewards.
> >
> >
> >
> > **Q2** When performing multiple rollouts on the same problem, does the frequency of code reasoning vary across samples, and if so, does this variation correlate with accuracy?
> >
> > **AQ2** We found that when a model chooses to use code reasoning, it typically generates code consistently for a given question across different samples. In Figures 27–29, we perform an interesting intervention by changing the narrative of the question. As a result, Qwen switches from code reasoning to natural language reasoning, and the answer can become incorrect after this switch (Figure 29), seemingly due to the change in the question’s narrative.
> >
> >
> > **Q3** Have the authors analyzed the quality or validity of the generated code (e.g., whether it is executable, syntactically correct, or error-free)? Is there a correlation between code quality and problem-solving performance, or does RLVR surface code-like reasoning patterns regardless of actual code validity?
> >
> > **AQ3** Yes! We have qualitatively validated some output examples and confirmed that the generated code is executable and that the answers following the code match the actual execution results. In all the examples we checked, the code was executable. Moreover, we also tested variants of the questions to see whether the model can still generate good code and predict the code outputs—as shown in Figures 24 and 25. Interestingly, when we change the narrative of the question, the model no longer generates code, and the answer accuracy appears to drop in the qualitatively checked examples, as shown in Figures 27 and 28. We conjecture that this may be related to data augmentation methods used during the Qwen model’s pretraining or mid-training.

---

### Author Response · Authors · 2025-12-04
**Thanks all for your time! We summarize reviewers' feedback and our responses here (1/2).**

Thank you to all reviewers for their invaluable feedback! We are glad all reviewers appreciated the expansiveness of our experiments and our timely finding that research should be tested on more than a single model (Qwen). We truly appreciate all reviewers’ time and effort.

We’d like to give special thanks to our AC for their extra time and effort in light of recent events. To help lighten the workload, we use this general response to summarize the reviewers’ feedback and our responses.

**Overall, feedback is polarized, with reviewer w4Pw in strong support** (“The paper presents a striking discovery… a valuable contribution that will benefit the broader research community”) and with **concerns raised by reviewers K7Z4 and duSx**, who asked about (1) the justification for our experimental design choices and (2) the interpretation and generality of our conclusions. We believe many of these concerns stem from a miscommunication of our intended claims, which we now seek to clarify. We also respond to each reviewers’ thoughtful questions directly in the responses below, and will revise our paper to make these points more clear.

### **Our key contribution**
We show and analyze a striking empirical fact: **some** Qwen models exhibit substantial improvements under RLVR on certain math tasks even with provably information-free spurious rewards. This result suggests that (1) current RLVR research practice—which often exclusively reports results on Qwen—may rely heavily on idiosyncrasies of the Qwen family, and (2) many widely reported RLVR gains at the time of submission may have arised not from better reward supervision but from unintended interactions with model behaviors from pretraining. We believe our work has already shaped how other researchers design their experiments as a direct consequence of this finding, for instance by testing on more model families.

### **Reviewer K7Z4’s main concern is why spurious rewards did not similarly improve performance in non-Qwen models.** In particular, given that spurious rewards can provably upweight existing behaviors through mode-collapse (Section 5), then “why don’t other base models also improve?” (reviewer K7Z4).

This concern was unfortunately raised as a result of miscommunication that we will make clearer in our revised manuscript. We clarify that our goal is **not** to advocate for using spurious rewards as an effective, general method for improving performance across tasks and models (L19-20, spurious rewards “lack [a] useful reward signal”) Instead, our paper presents a novel analytical approach that reveals the overlooked flaws in experimental designs from prior work, where—as noted above—researchers tend to draw generic conclusions from RLVR experiments on a single model family (for example, at the time of our submission, [2,3,4] draw conclusions from Qwen family models; other works listed in L242-250).

In particular, we conjecture that Qwen models have already internalized a strong mode of behavior for math reasoning during pretraining—potentially due to extensive code-style augmentation or accidental test-set contamination [1]. Under this intuition, spurious rewards do not “teach” new reasoning strategies but instead collapse the model’s distribution to more consistently adopt these pre-existing patterns. Thus, the fact that spurious rewards and the clipping bias from random rewards *do not* yield similar improvements in other model families is not a weakness of our analysis (**W2, W3 reviewer K7Z4**); rather, it is core evidence that RL training depends on modes learned during pretraining. Put in other words, our work contributes empirical and theoretical evidence that random rewards incentivize collapse onto the model’s existing modes. Whether or not that mode is useful (and hence whether or not spurious rewards yield gains) depends on what the model was pretrained on. To provide further evidence, we trace code reasoning as an illustrative example of one possible pre-existing mode that is (1) useful but *inconsistently used* by Qwen math models (i.e. only around half of the time, see Table 1) before RL training and (2) is upweighted over the course of RLVR training on spurious rewards.

---

> ### Author Response · Authors · 2025-12-04
> **Thanks all for your time! We summarize the reviewers' feedback and our responses here (2/2).**
>
> ### **Reviewer duSx raised concerns that this relationship between code reasoning frequency and downstream performance after spurious reward RLVR is not casually shown by our experiments—for example, other patterns may also play a role.**
>
> We apologize for the miscommunication, and would like to clarify that our goal is not to argue that increased code reasoning *causes all* gains from spurious reward training. Instead, we intend to discuss code reasoning primarily as an “illuminating case study” (L270) of evidence that RL can surface pre-existing useful behaviors, of which code reasoning is one. We directly acknowledge that other reasoning behaviors (such as no repetition, detailed in Appendix G) may also be upweighted. Indeed, a complete causal study of what pre-existing behaviors spurious rewards work through would require intervening at pretraining to remove the mode of code reasoning; hence showing causality is beyond the scope of our project. Fortunately, we are able to show (1) causal evidence that the mode of code reasoning behavior is useful to upweight (Section 4.3, also our individual response to reviewer duSx), and that (2) spurious rewards can upweight this mode even without any new information. For example, we show a casual analysis of how the clipping bias in GRPO incentivizes mode collapse even with random rewards (Section 5), which we are optimistic may help inform future RL algorithm design.
>
>
> Thank you again for your time and effort. We hope you find the above summarized discussion useful. Note that reviewers also raised further lower-level questions that we address directly in the comments below. We will revise our paper to clarify all points discussed in the next version of our paper. Thank you all for your invaluable help in strengthening our work!
>
> [1] Wu, Mingqi, et al. "Reasoning or memorization? unreliable results of reinforcement learning due to data contamination." arXiv preprint arXiv:2507.10532 (2025).
>
> [2] Zuo, Yuxin, et al. "Ttrl: Test-time reinforcement learning." arXiv preprint arXiv:2504.16084 (2025).
>
> [3] Wang, Yiping, et al. "Reinforcement learning for reasoning in large language models with one training example." arXiv preprint arXiv:2504.20571 (2025).
>
> [4] Zhao, Xuandong, et al. "Learning to reason without external rewards." arXiv preprint arXiv:2505.19590 (2025).

---

### Note · Program_Chairs · 2026-01-17
**Submission Desk Rejected by Program Chairs**

The following references in this submission do not refer to real documents and/or have major errors in bibliographic information:

 DeepSeek-Math. Deepseek-r1-zero: Advancing mathematical reasoning through step-by-step exploration. arXiv preprint arXiv:2404.01140, 2024.